# Uganda's Hydropower System Resilience to Extreme Climate Variability

**Francis Mujjuni** * , **Thomas Betts and Richard Blanchard**

Centre for Renewable Energy Systems Technology (CREST), Wolfson School of Mechanical, Electrical and Manufacturing Engineering, Epinal Way, Loughborough University, Loughborough LE11 3TU, UK; t.r.betts@lboro.ac.uk (T.B.); r.e.blanchard@lboro.ac.uk (R.B.)

\* Correspondence: f.mujjuni@lboro.ac.uk or francis.mujjuni@mak.ac.ug; Tel.: +44-7424-370334

**Abstract:** This study was motivated by the high reliance on hydropower plants (HPPs) developed and planned along the river Nile and the fact that drought events are the most imminent and drastic threats to Uganda's power production. The study aimed to assess HPPs' resilience and the effectiveness of selected adaptation measures. The climate, land, energy, and water system (CLEWs) framework was employed to assess resilience amidst competing water demands and stringent environmental flow requirements. Under extreme dry conditions, power generation could plummet by 91% over the next 40 years, which translates into an annual per capita consumption of 19 kWh, barely sufficient to sustain a decent socioeconomic livelihood. During arid conditions, climate models predicted an increase in streamflow with increasing radiative forcing. Restricting the ecological flow to 150 m³/s could improve generation by 207%. In addition, if planned power plants were to be built 5 years ahead of schedule, the normalized mean annual plant production could increase by 23%. In contrast, increasing reservoir volumes for planned power plants will have no significant impact on generation. The path to HPP resilience could entail a combination of diversifying the generation mix, installing generators with varying capacities, and incorporating adjustable orifices on reservoirs.

**Keywords:** resilience; hydropower; climate change; extreme events; drought; CLEWs; WEAP

## 1. Introduction

It is projected that Uganda will face immense pressures arising from the increased frequency, intensity, and variability of extreme weather events [1]. Flood and drought events have been particularly cited to be the most imminent and devastating for power systems [2]. In context, 92% of Uganda's electricity production comes from hydropower plants, 64% of the installed capacity of which is developed along the river Nile (R. Nile). The share of installed capacity will grow to 76% with the expected commissioning of the 600 MW Karuma hydropower project in 2023. In addition, the R. Nile is seen as a key source of water for supporting an ambitious irrigation plan [3] and provision of water for a rapidly growing population, which is projected to double by 2050.

Since 1967, Uganda has experienced about 10 major droughts [4]. In just five of those droughts (1987, 1998, 2002, 2005, and 2008), nearly 3.5 million people were affected and average annual losses of USD 20 million were incurred, which could rise to USD 200 million in the event of a one in 200-year drought [4,5]. It is estimated that 12% of the population is exposed to droughts elevating them to be the country's most potent natural disasters [1,6]. Historically, the occurrence of droughts mainly affected food production but droughts of 2004–2005 and 2010–2011 caused massive shortages in the R. Nile flows, plunging the country into a dire electricity supply deficit and leading to prolonged daily power cuts [1]. In response, the government negotiated and commissioned emergency thermal generators and the construction of Bujagali hydropower [7]. The current high end-user energy prices can be traced back to these two hasty decisions.

Several studies [8,9] in the past have assessed the impacts of climate change on the production of hydropower amidst competing water use interests in Uganda. They mostly relied on Phase 5 Coupled Model Intercomparison Project (CMIP5) reanalysis datasets. Since then, improved datasets, i.e., CMIP6, have been produced. In addition, previous assumptions and findings on demand, energy mix, energy prices, and plant production do not correspond to current observations. Moreover, limited emissions pathways were explored and most studies did not assess the effects of different adaptative measures on stream flow and electricity production. Therefore, this study aimed to quantify the effects of projected extreme climate variability on hydropower production and assess the probable gains in resilience emerging from selected adaptation measures.

This study contributes to the existing literature in various ways. This is the first study to use the entire CMIP6 database, which is available on a monthly scale to assess projected aridity and wetness within the Nile basin. The use of groundwater nodes to control the volume of L. Victoria and L. Kyoga is a novel application to mitigate against reanalysis data limitations. In addition, this is the first study employing the CLEW framework that calibrated the R. Nile's flow using L. Victoria levels and validated it across multiple gauging stations. Moreover, no other study known to the authors modeled as many as three adaptation scenarios as demonstrated in this paper.

## 2. Materials and Methods

This section details the methods implemented in this study. First, theoretical and graphical conceptual frameworks are presented followed by a description of data sources and processing. The section then describes the main modeling and simulation process accounting for the employed tools, modeling assumptions, and limitations. Finally, the section concludes with a description of how the model was calibrated and validated.

### 2.1. Conceptual Framework of Methodology

The core of the study entailed developing a water balance model (WBM) that accounts for the main hydrological features (lakes, and rivers), water demand centers (irrigation, and domestic and municipal usage), ecological flow requirements, and hydropower elements (plants and reservoirs) within the catchments of the large hydropower stations situated or planned along the R. Nile. Within the study area, there was a general lack of sufficient observational climate and hydrological data to support a comprehensive modeling process; therefore, the study leveraged data from General Circulation Models (GCMs) archived in the ERA5-Land and CMIP6 datasets [10].

A coupled water balance and hydropower model was implemented in four major phases: (1) the reference scenario from which climate data of the past (1981–2020) were cycled into the future (2021–2060), (2) the extreme scenario, which represented both the probable wettest and driest future climatic conditions, (3) the climate change scenario, which sought to quantify the effects of different emissions pathways on R. Nile discharge and hydropower production, and (4) the adaptation scenario, which evaluated the likely impacts of various adaptation measures on both flow and plants' generation. In all the phases, the model was subjected to similar water consumption priorities, demand rates, and supply arrangements. In general, the model appropriates precipitation and downstream inflows received within a catchment to evapotranspiration, groundwater recharge, and runoff. It is the latter that forms the bulk of river discharge from which a portion is used to generate power as explained in the subsequent sections. The detailed conceptual framework of the study can be seen in Figure 1.

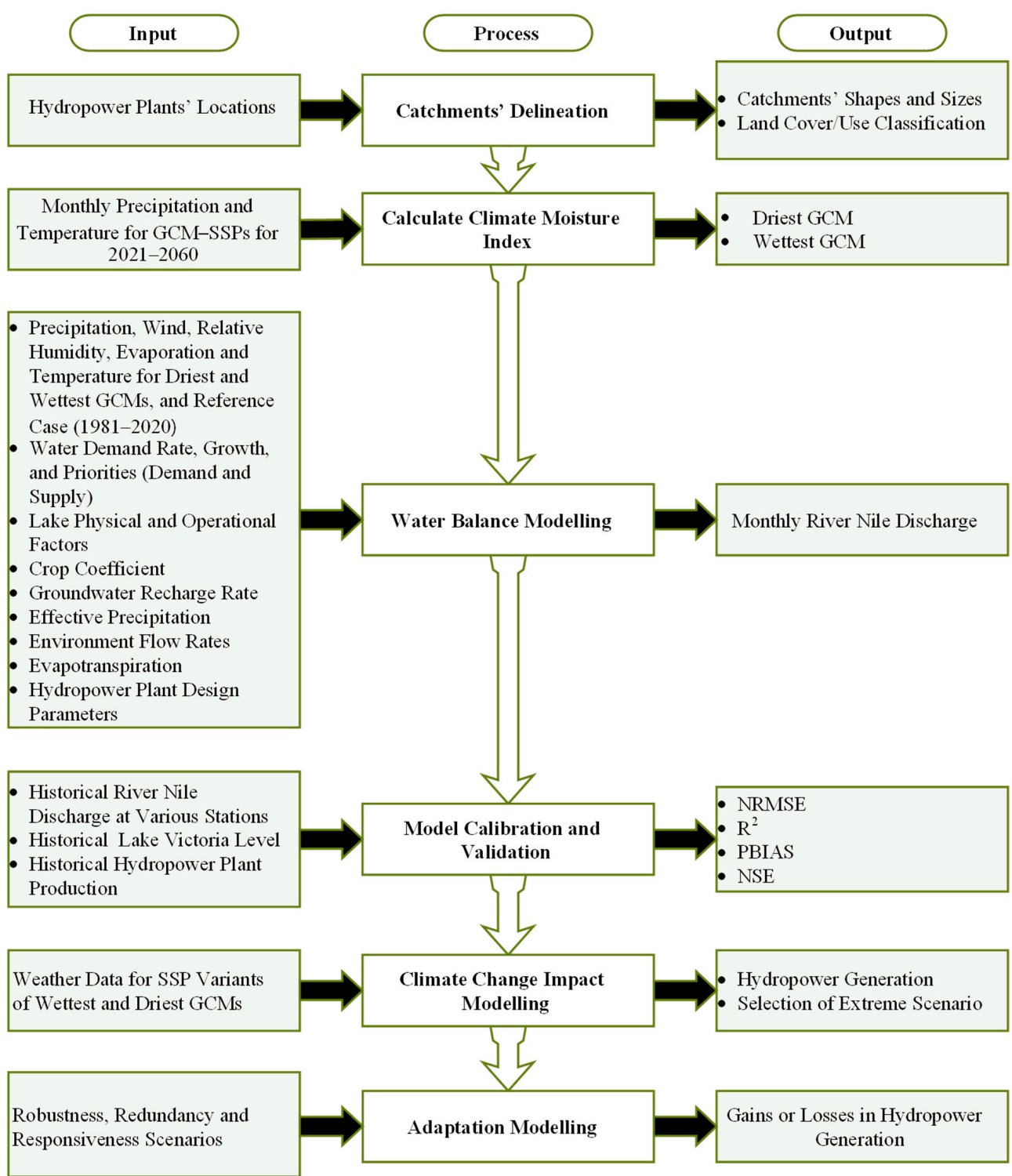

**Figure 1.** The Study Conceptual Framework.

## 2.2. Data Acquisition and Processing

Five data types were considered by this study: hydrology, hydropower, climate, land use and land cover, and demography data. The raw data files sources and relevance are listed in Table 1 and the processing of data is explained in subsequent Sections 2.2.1–2.2.5.

**Table 1.** Data Types, Sources, and Uses.

| Data Types | Parameters | Source | Use |
|---|---|---|---|
| Hydrology | • Lake Victoria levels | [11] | Calibration |
| | • River Victoria discharge | [11] | Validation |
| Large Hydropower Plants | • Digital Elevation Model | [12] | Catchments delineation |
| | • Hydropower plants design information | [13,14] | Reservoir modeling and hydropower generation |
| | • Hydropower generation | [15] | Validation |
| Climate | • Precipitation | | Evaluations of run-offs |
| | • Relative humidity | | |
| | • Radiation (shortwave and longwave) | | |
| | • Wind | [10] | Reference evapotranspiration |
| | • Pressure | | |
| | • Temperature | | |
| | • Dew Point | | |
| Land use | • Land cover maps | [16] | Land class disaggregation |
| Demography | • Population distribution | [17] | Water consumption |
| | • Water consumption | [18] | Water consumption |
| | • Irrigation rate and plan | [3,18] | Water consumption |

2.2.1. Hydrological Data

The bulk of the hydrological data were obtained from the Directorate of Water Resource Management (DWRM) [11]. The data consisted of historical measured flows of the R. Nile at four gauging stations (i.e., Lake Victoria outflow (Jinja Pier), Mbulambuti, Masindi Port, and Paara) and Lake Victoria (L. Victoria) water levels. The flow data files were composed of single daily readings whereas lake levels were recorded twice daily (at 0800 and 1600). Table 2 shows that a sizeable span of data were obtained for each of the gauging stations and, most noticeably, the data obtained at the L. Victoria outflow (hereafter also referred to as Owen Falls Dam or headrace) were nearly complete with only 2% missing both within the raw dataset and within the span, which formed the reference case of this study (1981–2020). The data were then averaged on a monthly scale. Several suspicious observations were made in the flow data for Mbulamuti, Masindi Port, and Paara, which included sudden rises and drops and several other values, which were considerably lower than those measured at the headrace of the river (see highlighted regions in Figure 2A) but the author's attempts to reconcile these observations with DWRM went unanswered. Therefore, the data were processed as obtained. Figure 2B,C, are scatter plots demonstrating the comparison of monthly flow data for the Jinja Pier gauging station with the other three stations. Accordingly, regression equations are presented consistent with the World Meteorological Organization reference periods (1961–1990 and 1991–2020).

Given the size of L. Victoria and L. Kyoga and their influence on the observed flow of the R. Nile, the model incorporated data on the physical and operational attributes of the two lakes. The data were obtained from various sources as seen in Table 3. The lakes' physical elements were characterized by inflows, storage capacity, net evaporation, and losses to groundwater whereas their operations were defined by the top of conservation, buffer, and inactive zones as well as the buffer coefficient. These characteristic elements of lakes are described in detail in [19].

**Table 2.** Key Information of the Hydrological Data Used in this Study. L. Victoria Outflow, R. Victoria Nile, and R. Kyoga Nile are Different Names for the Specific Location of R. Nile at the Headrace, between L. Victoria and L. Kyoga, and between L. Kyoga and L. Albert Respectively.

| Name | Data Type | Lat-Lon | Elevation (Masl (Metres above Sea Level)) | Catchment Area (km²) | Period | Missing Data (All) | Missing Data (1981–2020) |
|------|-----------|---------|--------------------------------------------|-----------------------|--------|--------------------|--------------------------|
| L. Victoria at Owen Falls Dam | Lake Level | 0.41, 33.21 | 1123 | 264,160 | 1948–2022 | 3% | 2% |
| L. Victoria at Jinja Pier | Lake Outflow | 0.41, 33.21 | 1123 | 264,160 | 1948–2022 | 3% | 2% |
| R. Victoria Nile at Mbulamuti | River Flow | 0.84, 33.03 | 1030 | 265,727 | 1956–2022 | 12% | 15% |
| R. Kyoga Nile at Masindi Port | River Flow | 1.69, 32.09 | 1021 | 338,465 | 1947–2020 | 21% | 28% |
| R. Kyoga Nile at Paraa | River Flow | 2.28, 31.56 | 641 | 349,207 | 1963–2021 | 55% | 49% |

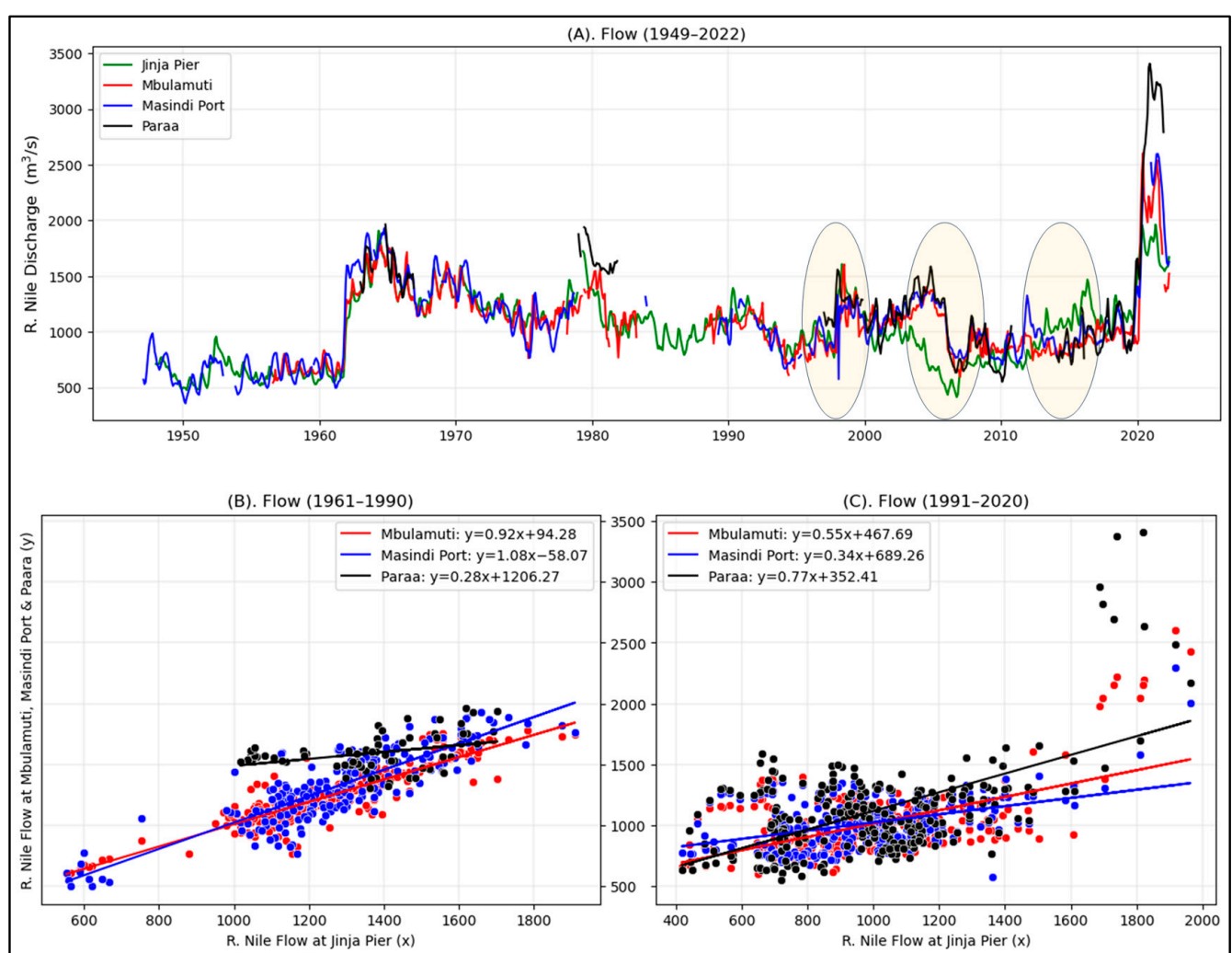

**Figure 2.** Monthly R. Nile Monthly Average Discharge at Selected Gauges: (**A**) is Plotted Raw Data with Observed Inconsistences Highlighted; (**B**,**C**) show Scatter Plots and Linear Regression Equations Comparing Data at Jinja Pier with the Other Three Gauging Sites.

**Table 3.** Physical and Operation Parameters for L. Victoria and L. Kyoga.

| Parameters | Units | L. Victoria | L. Kyoga | Data Source |
|---|---|---|---|---|
| Physical | | | | |
| • Instream flow | $m^3/s$ | Table 4 | | [9] |
| • Net Evaporation | mm | Evaporation–Precipitation | | |
| • Loss to Groundwater | % | Section 2.3.3 | | [9] |
| • Storage Capacity | MCM | 4,148,000 | 20,500 | [9,13,20] |
| • Initial Storage | MCM | 2,770,894 | 16,000 | [9] |
| • Volume Elevation Curve | MCM | $5 \times 10^{-13}H^2 + 2 \times 10^{-7}H + 7.88$ | $3 \times 10^{-3}H + 0.881$ | [20] |
| • Elevation (H) Range | m | 7.96–14 | 0–7.8 | [21–23] |
| Operational | | | | |
| • Top of Conservation | MCM | 3,128,000 | 16,000 | [9] |
| • Top of Buffer | MCM | 2,488,800 | 16,000 | [23,24] |
| • Top of Inactive | MCM | 2,000,000 | 0 | [23,24] |
| • Buffer Coefficient | | 0.5 | 1 | |

**Table 4.** Monthly Inflows in $m^3/s$ into Lakes Victoria and Kyoga.

| | Jan | Feb | Mar | Apr | May | Jun | Jul | Aug | Sep | Oct | Nov | Dec |
|---|---|---|---|---|---|---|---|---|---|---|---|---|
| L. Victoria | 638 | 531 | 744 | 1063 | 1275 | 744 | 531 | 638 | 638 | 531 | 850 | 1275 |
| L. Kyoga | 68 | 34 | 25 | 41 | 162 | 121 | 58 | 44 | 66 | 76 | 137 | 146 |

2.2.2. Large Power Plant Data

Uganda's power supply system is planned to have nearly 70 hydropower plants (HPPs) installed by 2040 [13,14,25]. In the context of this study, 10 were categorized as large hydropower plants based on their capacity exceeding 100 MW as seen in Table 5. Incidentally, all operational and planned large HPPs are located along the R. Nile. Currently, 22 HPPs with a capacity of 1000 MW are operational, 82% of which comprise large HPPs, which will rise to 90% (4000 MW) if all planned power plants are developed. In other words, the R. Nile's relevance, and contribution towards Uganda's energy supply will increase in the future.

**Table 5.** Operational State of Developed and Planned HPPs in Uganda.

| Category | Name of HPPs | Commercial Operational Date (COD) | State | Capacity (MW) |
|---|---|---|---|---|
| Large Hydro | Nalubaale | 1954 | Operational | 180 |
| | Kiira | 2004 | Operational | 200 |
| | Bujagali | 2012 | Operational | 250 |
| | Isimba | 2019 | Operational | 183 |
| | Karuma | 2023 | Construction | 600 |
| | Kiba | 2030 | Feasibility study | 400 |
| | Ayago | 2033 | Feasibility study | 600 |
| | Kalagala | 2035 | Feasibility study | 330 |
| | Oriang | 2037 | Feasibility study | 392 |
| | Murchison (Uhuru) | 2040 | Feasibility study | 655 |
| Small Hydros | Several plants | | Operational | 187 |
| | | | Licensed | 157 |
| | | | Feasibility study | 145 |

The HPP catchment areas were delineated from Aster Global Digital Elevation Model (DEM) Version-3 files obtained from the National Aeronautics and Space Administration's EarthData site [12]. The DEM files were downloaded as Tag Image File Format (.Tif) images within the bounding box of 26° W, 38° E, 6° N, and −7° S. The raw images were then merged into a single raster layer and processed into catchments (see Figure 3) using ArcMap 10.8.1 software.

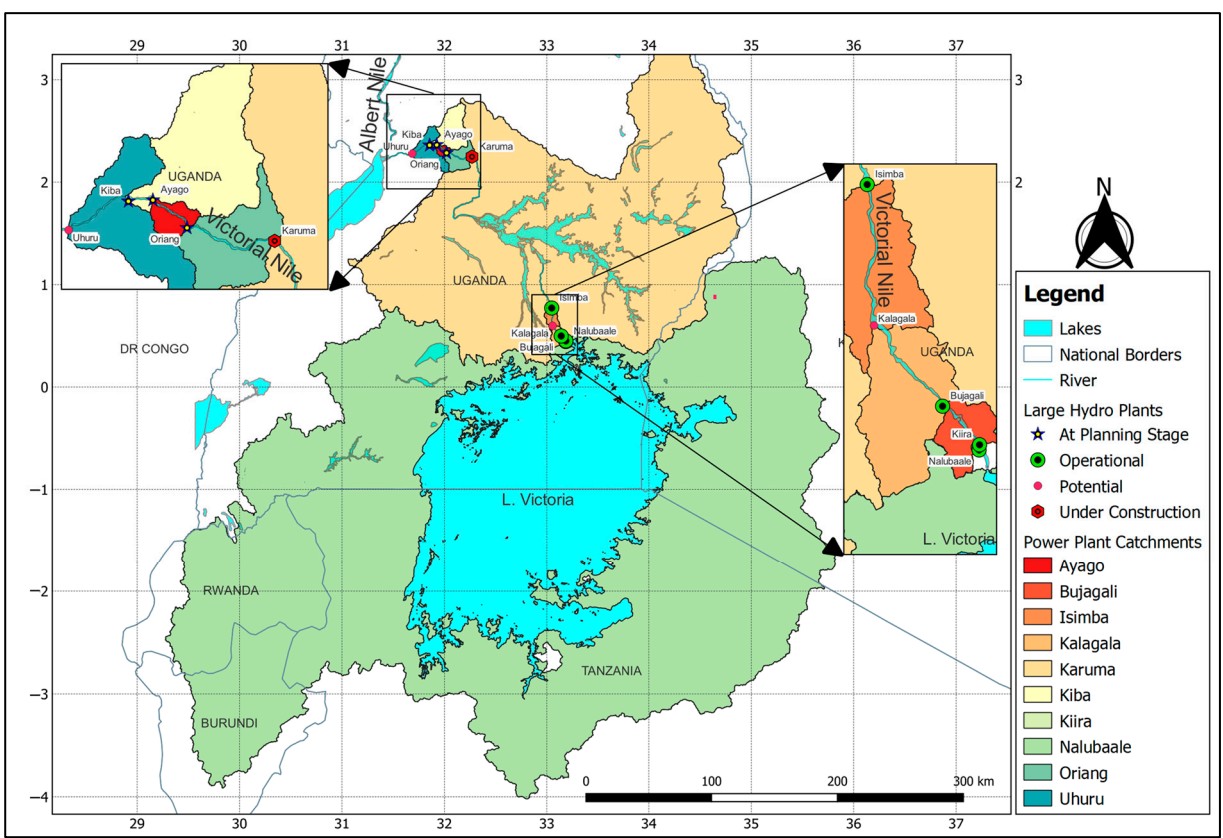

**Figure 3.** Catchments for Large Hydropower Plants.

The design data of the large HPPs as presented in Table 6 were obtained from various reports and studies [13,14,25–28]. All power plants can be classified as run-of-river given that they are essentially designed to operate with very little or no reliance on storage. Even Nalubaale and Kiira HPPs, which have L. Victoria as their reservoir, their intake flows are highly regulated by the dam operations; thus, they essentially can be considered run-of-river plants. The annual generation data recorded from 2011 to 2020 for the operational large HPPs (Nalubaale, Kiira, and Bujagali, Isimba) were obtained from the Electricity Regulatory Authority [29] This data were used to validate the output of the hydropower model.

**Table 6.** Large Hydropower Plants Salient Design Parameters.

| Parameters | Units | Nalubaale | Kiira | Bujagali | Isimba | Kalagala | Karuma | Oriang | Ayago | Kiba | Uhuru |
|---|---|---|---|---|---|---|---|---|---|---|---|
| Location | Lon, Lat | 33.2, 0.4 | 33.2, 0.5 | 33.1, 0.5 | 33.1, 0.8 | 33.1, 0.6 | 32.3, 2.3 | 32.1, 2.3 | 31.9, 2.4 | 31.9, 2.4 | 31.7, 2.3 |
| Design Head | m | 24 | 31 | 22 | 17.7 | 29 | 60 | 58 | 87 | 60 | 93 |
| Efficiency | % | 88 | 76 | 84 | 77 | 85 | 90 | 82 | 64 | 81 | 85 |
| Plant Factor | % | 62 | 38 | 100 | 65 | 64 | 65 | 81 | 81 | 82 | 41 |

**Table 6.** *Cont.*

| Parameters | Units | Nalubaale | Kiira | Bujagali | Isimba | Kalagala | Karuma | Oriang | Ayago | Kiba | Uhuru |
|---|---|---|---|---|---|---|---|---|---|---|---|
| Design Flow | m$^3$/s | 865 | 865 | 1375 | 1375 | 1375 | 1128 | 840 | 1100 | 840 | 840 |
| Reservoir Level | masl | 1135 | 1135 | 1112 | 1059 | 1088 | 1029 | N/A (Not Applicable) | N/A | N/A | 718 |
| Tail Water Level | masl | 1132 | 1126 | 1090 | 1041 | 1059 | 969 | N/A | N/A | N/A | 625 |
| Gross Storage | MCM | Lake Victoria | | 54 | 171 | 29 | 80 | N/A | N/A | N/A | 19 |

### 2.2.3. Climate Data

The analysis was undertaken at a catchment scale. Such spatial resolution, as indicated by the catchment areas in Figure 3, could not be realistically represented by ground measurements from weather gauges sparsely spread across the basin. In any case, the authors reached out to Uganda National Meteorological Authority (UNMA) [30], the body in charge of recording, archiving, and distributing climate data, yet for several months, no response was given regarding the request of obtaining or purchasing the required data. Several studies [8,9,21,22] in the past faced similar problems, and instead, they used reanalysis data.

This study improved upon previous studies by using newer and a more expansive catalog of GCMs. Monthly precipitation, temperature, relative humidity, wind, evaporation, and dewpoint data from the fifth generation of ECMWF (European Centre for Medium-Range Weather Forecasts) atmospheric reanalysis of the global climate (ERA5) and the sixth phase of the Coupled Model Intercomparison Project (CMIP6) were obtained from [10] and processed on a catchment scale. All grids from reanalysis data files that were within or intersected a given catchment boundary were averaged at every timestep. Figure 4A,B demonstrates an example of ERA5 data grids for wind and temperature within the Karuma HPP catchment, which were averaged, and the respective values assigned to the January 1981 timestep. Similarly, monthly precipitation and evaporation data for the two large lakes, L. Victoria and L. Kyoga, were processed by obtaining the average value of all grids touching or within lake boundaries as seen in Figure 5a,b. This study acquired the ERA5-Land monthly averaged reanalysis data with a latitude–longitude resolution of 0.1° × 0.1° (9 km) spanning a period of 1950–2020. These data have been demonstrated to be relatively more accurate and certainly of high spatial resolution [31,32] than the ERA interim or the Princeton Land Surface Hydrology Research Group, which was employed in previous studies [8,21,22].

In addition, monthly precipitation, wind, temperature, and relative humidity data for 58 GCMs within the CMIP6 database were accessed and processed. As seen in Table 7 below, the obtained GCM data were classified into five Shared Socioeconomic Pathway future emission scenarios (SSP1, SSP2, SSP3, SSP4, and SSP5), which indicate the different global changes and how they might affect future emissions. Different emissions scenarios are considered using Representative Concentration Pathways (RCPs), which are indicative of expected radiative forcing by the year 2100 in comparison to the 1750 baseline. This study acquired data for all RCPs (1.9, 2.6, 3.4, 3.4OS, 4.5, 7.0, and 8.5 W m$^{-2}$), which were retrievable from the ECMWF database at a monthly scale. The relationship between RCP and expected warming is presented in Table 8. For a detailed explanation of the historical context, motivation, and data structure of the CMIP6 refer to O'Neill et al. [33] and Riahi et al. [34]. This study acquired, analyzed, and evaluated data from 181 GCM-SSP combinations. For example, in Table 7, it can be observed that ACCESS-CM2 had relevant data in SSP1-2.6, SSP2-4.5, SSP3-7.0, and SSP5-8.5. That is, one GCM was considered in four different emission pathways.

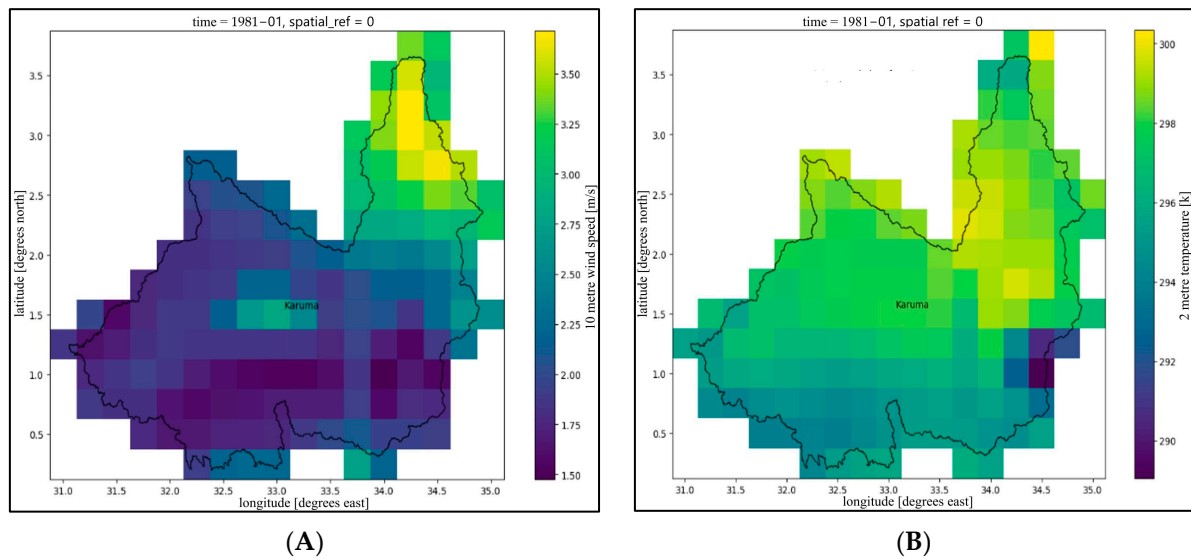

**Figure 4.** ERA5 Data Grids for Karuma HPP Catchment for January 1981: (**A**) is Wind Speeds in m/s at 10 m and (**B**) is Temperature in K at 2 m.

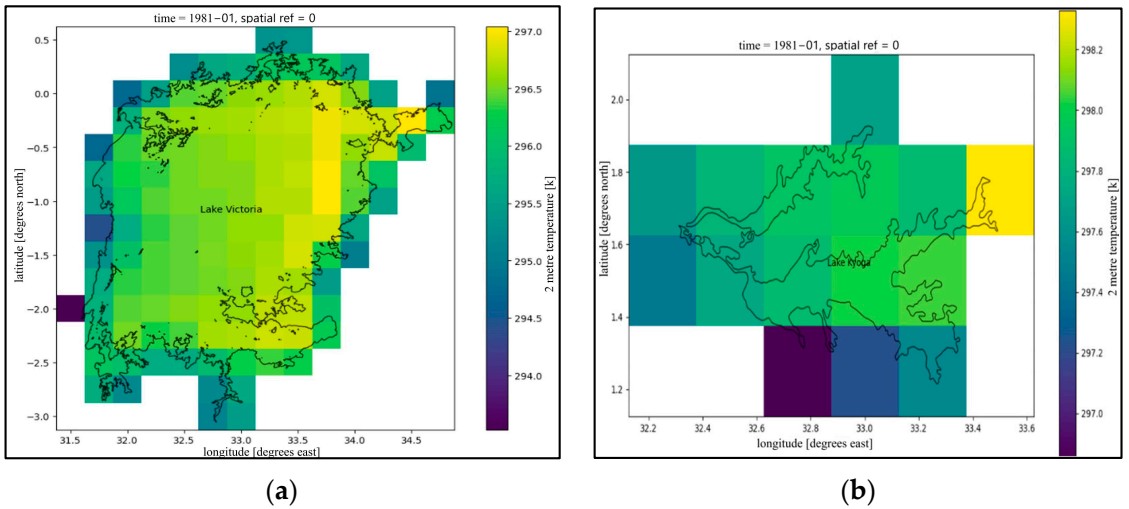

**Figure 5.** ERA5 Data Grids for Lake Victoria (**a**) and Lake Kyoga (**b**) for Temperature in K for January 1981.

**Table 7.** Data Obtained from GCMs in CMIP6 and ERA5-Land Databases. In total 629 Datasets are Represented in this Table from 181 GCM-SSP Combinations (P—Precipitation, W—Wind Speed, T—Temperature, R—Relative Humidity).

| GCMs | Historical | | | | SSP1 1.9 | | | | 2.6 | | | | SSP2 4.5 | | | | SSP3 7.0 | | | | SSP4 3.4 | | | | SSP5 3.4OS | | | | 8.5 | | | | Resolution |
|---|---|---|---|---|---|---|---|---|---|---|---|---|---|---|---|---|---|---|---|---|---|---|---|---|---|---|---|---|---|---|---|---|---|
| | P | W | R | T | P | W | R | T | P | W | R | T | P | W | R | T | P | W | R | T | P | W | R | T | P | W | R | T | P | W | R | T | |
| ERA5-Land | ✓ | ✓ | ✓ | ✓ | | | | | | | | | | | | | | | | | | | | | | | | | | | | | 9 km |
| ACCESS-CM2 | | | | | | | | | ✓ | ✓ | ✓ | ✓ | ✓ | ✓ | ✓ | ✓ | ✓ | ✓ | ✓ | ✓ | | | | | | | | | ✓ | ✓ | ✓ | ✓ | 250 km |
| ACCESS-ESM1-5 | | | | | | | | | | | | | | | | | | | | | | | | | | | | | | | | | 250 km |
| AWI-CM-1-1-MR | | | | | | | | | ✓ | ✓ | ✓ | ✓ | ✓ | ✓ | ✓ | ✓ | ✓ | ✓ | ✓ | ✓ | | | | | | | | | ✓ | ✓ | ✓ | ✓ | 100 km |

**Table 7.** *Cont.*

| GCMs | Historical | | | | SSP1 1.9 | | | | SSP1 2.6 | | | | SSP2 4.5 | | | | SSP3 7.0 | | | | SSP4 3.4 | | | | SSP5 3.4OS | | | | SSP5 8.5 | | | | Resolution |
|---|---|---|---|---|---|---|---|---|---|---|---|---|---|---|---|---|---|---|---|---|---|---|---|---|---|---|---|---|---|---|---|---|---|
| | P | W | R | T | P | W | R | T | P | W | R | T | P | W | R | T | P | W | R | T | P | W | R | T | P | W | R | T | P | W | R | T | |
| AWI-ESM-1-1-LR | | | | | | | | | | | | | | | | | | | | | | | | | | | | | | | | | 250 km |
| BCC-CSM2-MR | | | | | | | | | ✓ | ✓ | ✓ | ✓ | ✓ | ✓ | ✓ | ✓ | ✓ | ✓ | ✓ | ✓ | | | | | | | | | ✓ | ✓ | | ✓ | 100 km |
| BCC-ESM1 | | | | | | | | | | | | | | | | | | | | | | | | | | | | | | | | | 250 km |
| CAMS-CSM1-0 | | | | | ✓ | | | | ✓ | ✓ | | | ✓ | ✓ | | | ✓ | ✓ | | | ✓ | | | | ✓ | | | | | | | ✓ | 100 km |
| CanESM5 | ✓ | ✓ | ✓ | ✓ | ✓ | ✓ | ✓ | ✓ | ✓ | | | | | | | | | | | | ✓ | ✓ | ✓ | ✓ | ✓ | ✓ | ✓ | ✓ | | | | ✓ | 500 km |
| CanESM5-CanOE | | | | | | | | | ✓ | | | | ✓ | ✓ | ✓ | ✓ | ✓ | ✓ | ✓ | ✓ | ✓ | | | | | | | | ✓ | ✓ | ✓ | ✓ | 500 km |
| CESM2 | | | | | | | | | ✓ | ✓ | ✓ | ✓ | ✓ | ✓ | ✓ | ✓ | ✓ | ✓ | ✓ | ✓ | | | | | | | | | ✓ | ✓ | ✓ | ✓ | 100 km |
| CESM2-FV2 | | | | | | | | | | | | | | | | | | | | | | | | | | | | | | | | | 250 km |
| CESM2-WACCM | | | | | | | | | | | | | | | | | ✓ | ✓ | ✓ | ✓ | | | | | ✓ | ✓ | ✓ | | ✓ | | | ✓ | 100 km |
| CESM2-WACCM-FV2 | | | | | | | | | | | | | | | | | | | | | | | | | | | | | | | | | 100 km |
| CIESM | | | | | | | | | ✓ | ✓ | | | ✓ | ✓ | | | | | | | | | | | | | | | | | ✓ | ✓ | 100 km |
| CMCC-CM2-HR4 | | | | | | | | | | | | | | | | | | | | | | | | | | | | | ✓ | | | | 100 km |
| CMCC-CM2-SR5 | | | | | | | | | ✓ | ✓ | ✓ | ✓ | ✓ | ✓ | ✓ | ✓ | ✓ | ✓ | ✓ | ✓ | | | | | | | | | ✓ | ✓ | ✓ | ✓ | 100 km |
| CMCC-ESM2 | | | | | | | | | ✓ | ✓ | ✓ | ✓ | ✓ | ✓ | ✓ | ✓ | | | | | | | | | | | | | ✓ | ✓ | ✓ | ✓ | 100 km |
| CNRM-CM6-1 | | | | | | | | | ✓ | ✓ | ✓ | ✓ | ✓ | ✓ | ✓ | ✓ | ✓ | ✓ | | | | | | | | | | | ✓ | ✓ | ✓ | ✓ | 250 km |
| CNRM-CM6-1-HR | | | | | | | | | ✓ | ✓ | ✓ | ✓ | ✓ | ✓ | ✓ | ✓ | ✓ | ✓ | ✓ | ✓ | | | | | | | | | ✓ | ✓ | ✓ | ✓ | 50 km |
| CNRM-ESM2-1 | ✓ | ✓ | ✓ | ✓ | ✓ | ✓ | ✓ | ✓ | ✓ | ✓ | ✓ | ✓ | ✓ | ✓ | ✓ | ✓ | ✓ | ✓ | ✓ | ✓ | ✓ | ✓ | ✓ | ✓ | ✓ | ✓ | ✓ | ✓ | ✓ | ✓ | ✓ | ✓ | 250 km |
| E3SM-1-0 | | | | | | | | | | | | | | | | | | | | | | | | | | | | | | | | | 100 km |
| E3SM-1-1 | | | | | | | | | | | | | | | | | | | | | | | | | | | | | ✓ | ✓ | | ✓ | 100 km |
| E3SM-1-1-ECA | | | | | | | | | | | | | | | | | | | | | | | | | | | | | | | | | 100 km |
| EC-Earth3 | ✓ | ✓ | ✓ | ✓ | | | | | | | | | | | | | | | | | ✓ | ✓ | | | ✓ | ✓ | ✓ | | ✓ | | | | 100 km |
| EC-Earth3-AerChem | | | | | | | | | | | | | | | | | ✓ | | | | | ✓ | ✓ | | | | | | | | | | 100 km |
| EC-Earth3-CC | | | | | | | | | | ✓ | | | ✓ | ✓ | ✓ | ✓ | | | | | | | | | | | | | ✓ | ✓ | ✓ | ✓ | 100 km |
| EC-Earth3-Veg | ✓ | ✓ | | | | | | | ✓ | | | | | | | | | | | | | | | | | | | | | | | | 100 km |
| EC-Earth3-Veg-LR | ✓ | ✓ | ✓ | ✓ | ✓ | ✓ | ✓ | ✓ | ✓ | ✓ | ✓ | ✓ | ✓ | ✓ | ✓ | ✓ | ✓ | ✓ | ✓ | ✓ | | | | | | | | | ✓ | ✓ | ✓ | ✓ | 250 km |
| FGOALS-f3-L | | | | | | | | | ✓ | ✓ | | | ✓ | ✓ | ✓ | | ✓ | ✓ | ✓ | | ✓ | | | | | | | | ✓ | ✓ | | ✓ | 100 km |
| FGOALS-g3 | | | | | ✓ | ✓ | | | ✓ | ✓ | ✓ | ✓ | ✓ | ✓ | ✓ | ✓ | ✓ | ✓ | ✓ | ✓ | ✓ | ✓ | ✓ | | ✓ | ✓ | | | ✓ | ✓ | ✓ | ✓ | 250 km |
| FIO-ESM-2-0 | | | | | | | | | ✓ | ✓ | ✓ | ✓ | ✓ | ✓ | ✓ | ✓ | | | | | | | | | | | | | ✓ | ✓ | ✓ | ✓ | 100 km |
| GFDL-ESM4 | ✓ | ✓ | ✓ | ✓ | | | | | ✓ | ✓ | ✓ | ✓ | ✓ | ✓ | ✓ | ✓ | ✓ | ✓ | ✓ | | | | | | | | | | ✓ | ✓ | ✓ | ✓ | 100 km |
| GISS-E2-1-G | ✓ | ✓ | | | | | | | ✓ | | | | | | | | | | | | ✓ | ✓ | | | ✓ | | | | | | | | 250 km |
| GISS-E2-1-H | | | | | | | | | | | | | | | | | | | | | | | | | | | | | | | | | 250 km |
| HadGEM3-GC31-LL | | | | | | | | | ✓ | ✓ | ✓ | ✓ | ✓ | ✓ | ✓ | | | | | | | | | | | | | | ✓ | ✓ | ✓ | ✓ | 250 km |
| HadGEM3-GC31-MM | | | | | | | | | ✓ | ✓ | ✓ | ✓ | | | | | | | | | | | | | | | | | ✓ | ✓ | ✓ | ✓ | 100 km |
| IITM-ESM | | | | | | | | | ✓ | ✓ | | | ✓ | ✓ | ✓ | | ✓ | ✓ | ✓ | | ✓ | | | | | | | | ✓ | ✓ | | ✓ | 250 km |
| INM-CM4-8 | | | | | | | | | ✓ | ✓ | | | ✓ | ✓ | ✓ | | ✓ | ✓ | ✓ | | ✓ | | | | | | | | ✓ | ✓ | ✓ | ✓ | 100 km |
| INM-CM5-0 | | | | | | | | | ✓ | ✓ | | | ✓ | ✓ | ✓ | ✓ | ✓ | ✓ | ✓ | ✓ | ✓ | ✓ | | | | | | | ✓ | ✓ | ✓ | ✓ | 100 km |
| IPSL-CM5A2-INCA | | | | | | | | | ✓ | ✓ | | | ✓ | | | | ✓ | ✓ | ✓ | | ✓ | | | | | | | | ✓ | | | | 500 km |
| IPSL-CM6A-LR | ✓ | ✓ | ✓ | ✓ | ✓ | ✓ | ✓ | ✓ | ✓ | ✓ | ✓ | ✓ | ✓ | ✓ | ✓ | ✓ | ✓ | ✓ | ✓ | ✓ | ✓ | ✓ | ✓ | ✓ | ✓ | ✓ | ✓ | ✓ | ✓ | ✓ | ✓ | ✓ | 250 km |
| KACE-1-0-G | | | | | | | | | | ✓ | | | ✓ | ✓ | ✓ | ✓ | ✓ | ✓ | ✓ | ✓ | ✓ | ✓ | | | | | | | ✓ | ✓ | ✓ | ✓ | 250 km |
| KIOST-ESM | | | | | | | | | ✓ | ✓ | ✓ | | ✓ | ✓ | ✓ | | | | | | | | | | | | | | ✓ | ✓ | | ✓ | 250 km |

**Table 7.** *Cont.*

| GCMs | Historical | | | | SSP1 | | | | | | | | SSP2 | | | | SSP3 | | | | SSP4 | | | | SSP5 | | | | | | | | Resolution |
|---|---|---|---|---|---|---|---|---|---|---|---|---|---|---|---|---|---|---|---|---|---|---|---|---|---|---|---|---|---|---|---|---|---|
| | | | | | 1.9 | | | | 2.6 | | | | 4.5 | | | | 7.0 | | | | 3.4 | | | | 3.4OS | | | | 8.5 | | | | |
| | P | W | R | T | P | W | R | T | P | W | R | T | P | W | R | T | P | W | R | T | P | W | R | T | P | W | R | T | P | W | R | T | |
| MCM-UA-1-0 | | | | | | | | | | | | ✓ | ✓ | ✓ | | | ✓ | ✓ | | | ✓ | | | | | | | | ✓ | | | ✓ | 250 km |
| MIROC6 | | | | | ✓ | ✓ | | | ✓ | ✓ | | ✓ | ✓ | ✓ | | ✓ | ✓ | ✓ | | ✓ | ✓ | ✓ | | ✓ | ✓ | ✓ | | ✓ | ✓ | ✓ | ✓ | ✓ | 250 km |
| MIROC-ES2H | | | | | | | | | | | | | | | | | | | | | | | | | | | | | | | | | 250 km |
| MIROC-ES2L | ✓ | ✓ | ✓ | ✓ | ✓ | ✓ | ✓ | ✓ | ✓ | ✓ | ✓ | ✓ | ✓ | ✓ | ✓ | ✓ | | | | | | | | | ✓ | ✓ | ✓ | ✓ | ✓ | ✓ | ✓ | ✓ | 500 km |
| MPI-ESM-1-2-HAM | | | | | | | | | | | | | | | | | ✓ | ✓ | ✓ | ✓ | | | | | | | | | | | | | 250 km |
| MPI-ESM1-2-HR | | | | | | | | | | | | | | | | | | | | | | | | | | | | | | | | | 100 km |
| MPI-ESM1-2-LR | | | | | | | | | ✓ | ✓ | ✓ | ✓ | ✓ | ✓ | | | ✓ | ✓ | ✓ | ✓ | | | | | | | | | ✓ | ✓ | ✓ | ✓ | 250 km |
| MRI-ESM2-0 | ✓ | ✓ | | | ✓ | ✓ | | ✓ | ✓ | ✓ | | ✓ | ✓ | ✓ | | ✓ | ✓ | ✓ | | ✓ | ✓ | ✓ | | ✓ | ✓ | ✓ | | ✓ | ✓ | ✓ | ✓ | ✓ | 100 km |
| NESM3 | | | | | | | | | ✓ | ✓ | ✓ | ✓ | ✓ | ✓ | ✓ | ✓ | | | | | | | | | | | | | ✓ | ✓ | ✓ | ✓ | 250 km |
| NorCPM1 | | | | | | | | | | | | | | | | | | | | | | | | | | | | | | | | | 250 km |
| NorESM2-LM | | | | | | | | | ✓ | ✓ | | | ✓ | ✓ | | ✓ | ✓ | ✓ | | ✓ | ✓ | | | | | | | | ✓ | ✓ | | ✓ | 250 km |
| NorESM2-MM | | | | | | | | | ✓ | ✓ | | | ✓ | ✓ | | ✓ | ✓ | ✓ | | ✓ | ✓ | | | | | | | | ✓ | ✓ | | ✓ | 100 km |
| SAM0-UNICON | | | | | | | | | | | | | | | | | | | | | | | | | | | | | | | | | 100 km |
| TaiESM1 | | | | | | | | | ✓ | ✓ | ✓ | ✓ | ✓ | ✓ | ✓ | ✓ | ✓ | ✓ | ✓ | ✓ | | | | | | | | | ✓ | ✓ | ✓ | ✓ | 100 km |
| UKESM1-0-LL | ✓ | ✓ | ✓ | ✓ | ✓ | ✓ | ✓ | ✓ | ✓ | ✓ | ✓ | ✓ | ✓ | ✓ | ✓ | ✓ | ✓ | ✓ | ✓ | ✓ | ✓ | ✓ | ✓ | ✓ | ✓ | ✓ | ✓ | ✓ | ✓ | ✓ | ✓ | ✓ | 250 km |

**Table 8.** Relationship between Representative Concentration Pathways and Expected Temperature Increase.

| RCPs (W m$^{-2}$) | Radiative Forcing Category | Expected Warming (°C) by 2100 |
|---|---|---|
| 1.9 | Low | 1.4 |
| 2.6 | Low | 1.8 |
| 3.4 | Low | 2.2 |
| 3.4OS | Overshoot | 2.3 |
| 4.5 | Medium | 2.7 |
| 7.0 | High | 3.6 |
| 8.5 | High | 4.4 |

Source of data [33–35].

### 2.2.4. Land Cover Data

Land cover classification data were obtained from European Space Agency-Climate Change Initiative [16]. The data for 2020 were selected and used to represent all years under consideration. In its raw form, the data contain 22 land classes as defined by the United Nations Food and Agriculture Organization's Land Cover Classification System and are archived at a horizontal resolution of 300 m. In this study, the classes were condensed into 8, namely, agriculture, forest, grassland, wetland, urban, shrubland, barren or sparse vegetation, and open water as seen in Figure 6. The area of each class within the catchments (as shown in Table 9) was computed and used as input into the WBM. The catchment areas in Table 9 represent the area in between two adjacent power plants as required input of the WBM, but the actual catchment area of each power plant is cumulative from left to right. That is, Uhuru's actual catchment area is 351,988 km$^2$ (summing all catchment areas from Nalubaale to Uhuru) and not just 875 km$^2$, which is the area between Kiba and Uhuru's planned Intakes.

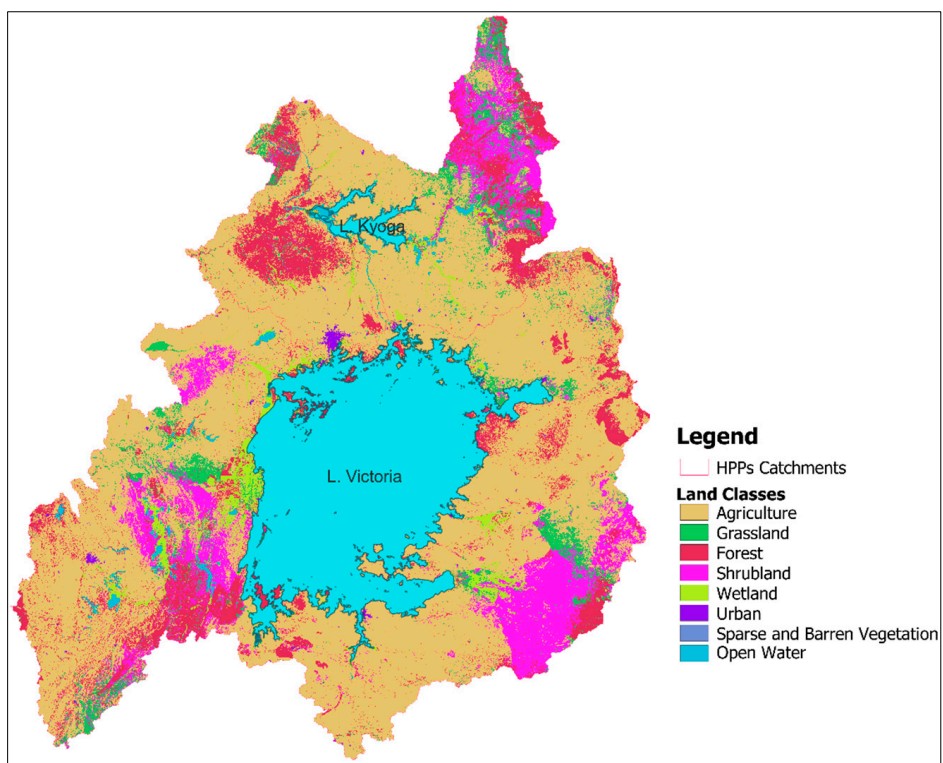

**Figure 6.** Land Cover Classes Within the Study Area.

**Table 9.** Share of Land Classes within Large Hydropower Plant Catchment Areas as Implemented in the Water Balance Model.

| | Units | Nalubaale | Kiira | Bujagali | Isimba | Kalagala | Karuma | Oriang | Ayago | Kiba | Uhuru |
|---|---|---|---|---|---|---|---|---|---|---|---|
| Catchment Area | km$^2$ | 266,078 | 0.63 | 61 | 248 | 218 | 82,431 | 679 | 135 | 1262 | 875 |
| 1. Agriculture | Share | 50.9% | 3% | 81.7% | 94.4% | 77.6% | 62.0% | 25.7% | 52.5% | 27.9% | 40.2% |
| 2. Forest | Share | 8.7% | 0% | 6.0% | 0.4% | 18.9% | 16.4% | 52.8% | 19.6% | 59.8% | 22.7% |
| 3. Grassland | Share | 2.3% | 0% | 0.0% | 0.0% | 0.0% | 3.7% | 13.3% | 14.5% | 9.1% | 29.6% |
| 4. Wetland | Share | 2.9% | 37% | 2.9% | 2.6% | 0.5% | 2.0% | 0.0% | 1.0% | 0.1% | 0.4% |
| 5. Urban | Share | 0.2% | 43% | 4.4% | 0.1% | 0.2% | 0.3% | 0.0% | 0.0% | 0.0% | 0.0% |
| 6. Shrubland | Share | 8.9% | 0% | 0.1% | 0.0% | 0.0% | 11.5% | 5.2% | 6.9% | 2.7% | 6.6% |
| 7. Barren Vegetation | Share | 0.0% | 0% | 0.0% | 0.0% | 0.0% | 0.0% | 0.0% | 0.0% | 0.0% | 0.0% |
| 8. Open Water | Share | 26.1% | 17% | 4.9% | 2.4% | 2.8% | 4.2% | 2.8% | 5.6% | 0.3% | 0.5% |

2.2.5. Demographic Data

The population within each catchment was computed from the census and household demography survey data obtained from the Uganda Bureau of Statistics (UBOS) [17]. Individual catchment populations were computed by aggregating district data bounded within a particular catchment area. In instances where a district extended to multiple catchments, the population was distributed equally amongst the relevant catchments. Given that the population data were used to account for water demand and usage within Uganda, the population outside Uganda but within the catchment areas was not considered. Figure 7 shows the population at the district scale for 2020 and illustrates its aggregation at a catchment scale. In the recent record of censuses, from 1991 to 2014, the annual population growth was 3.03%. This rate was used to estimate the annual population spanning 1981–2060.

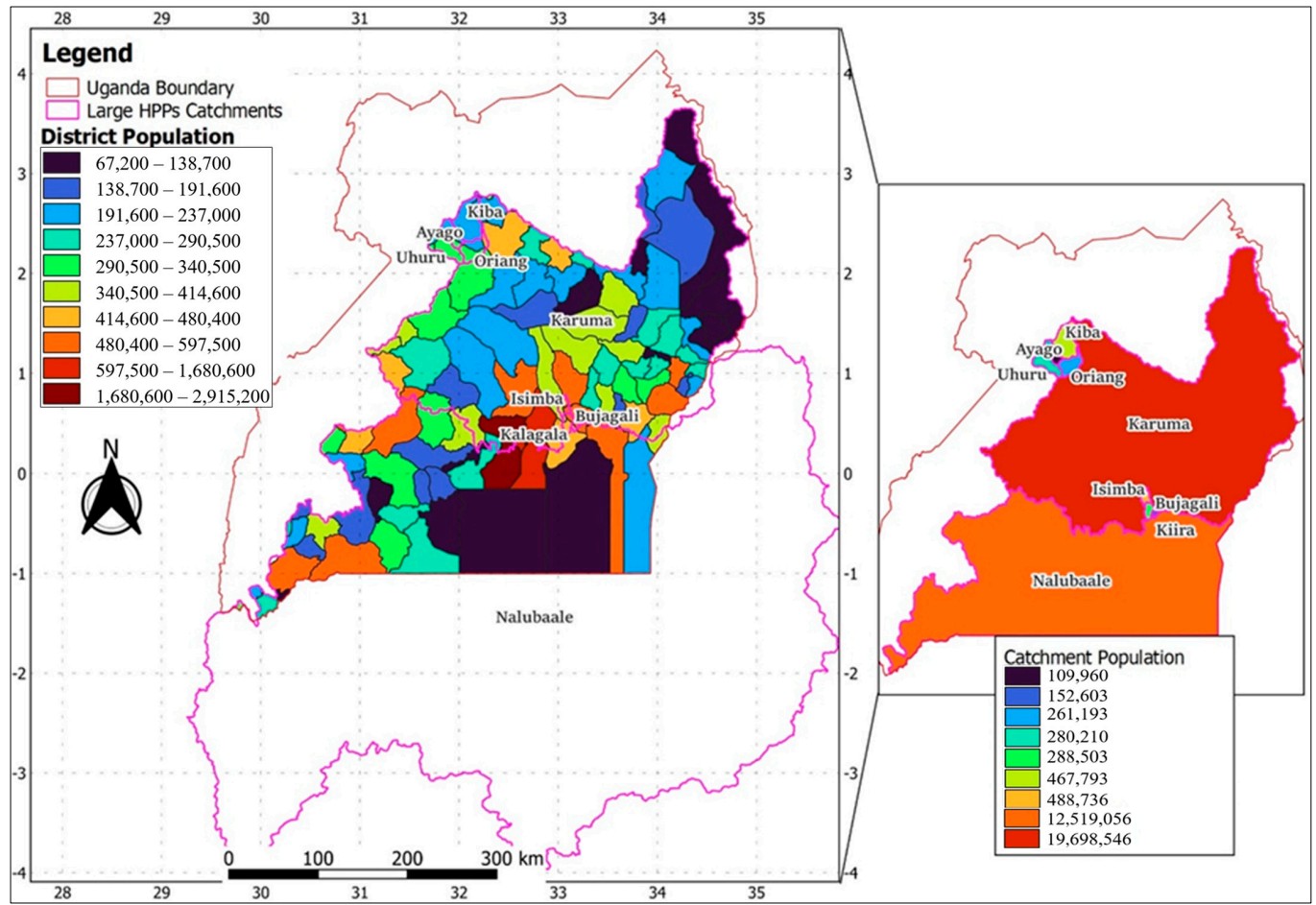

**Figure 7.** Catchment Population for 2020.

To account for the major drivers of water demand within the catchment, the study acquired and analyzed irrigation, domestic, and industrial water supply and consumption from the water accounts report published by UBOS [18]. The report covered the period from 2017 to 2020. Time series data were generated using 2020 as a baseline (captured in Table 10) with the following assumptions:

- The irrigated land area was considered to be 0.15% of the catchment agricultural area within Uganda. This was inferred from the 2020 data, in which the total irrigated area (approximately 100 km$^2$) within Uganda was 0.15% of the country's arable land area. The irrigated area for 2020 was considered to represent the period 1981–2020, whereas for 2021–2040, the study incorporated plans from the National Irrigation Policy [3], which proposed increasing irrigable land area to 15,000 km$^2$ by 2040. This plan infers an annual growth of 28.5% of irrigable land between 2020 and 2040. Given that there are no plans beyond 2040, it was considered that from 2041 to 2060, the growth rate would be 3% reflecting the projected population growth.
- The irrigation water demand was considered to be 2.6 million cubic meters per km$^2$ per year. The study considered that 50% of this was consumed or lost by the crops through evapotranspiration and the rest went to groundwater recharge.
- Municipal water demand, which aggregated both domestic and industrial water usage, was computed as 36,565 m$^3$ per person per year. The subsequent consumption was considered to be 2% of the demand and was estimated to grow at an annual rate of 2.24%.

**Table 10.** Population and Water Use Rates for 2020.

| Name | Population | Agricultural Land (km²) | Irrigated Land (km²) | Water Use Municipal (MCM) | Water Use Irrigation (MCM) |
|---|---|---|---|---|---|
| Ayago | 109,960 | 71 | 0.02 | 4021 | 0.044 |
| Bujagali | 152,602 | 55 | 0.05 | 5580 | 0.103 |
| Isimba | 488,735 | 244 | 0.33 | 17,870 | 0.622 |
| Kalagala | 288,502 | 211 | 0.18 | 10,549 | 0.351 |
| Karuma | 19,698,546 | 77,823 | 58.13 | 657,394 | 110.809 |
| Kiba | 397,251 | 1251 | 0.16 | 14,525 | 0.308 |
| Kiira | 152,602 | 1 | 0.00 | 5580 | 0.000 |
| Nalubaale | 12,519,056 | 30,666 | 20.33 | 367,597 | 38.754 |
| Oriang | 223,385 | 645 | 2.92 | 8168 | 0.286 |
| Uhuru | 280,210 | 855 | 0.15 | 10,246 | 0.667 |

*2.3. Water Balance Model*

This study was based on the CLEW (Climate, Land, Environment, and Water system) framework [36], which has been widely employed in recent studies [9,37,38] to holistically analyze and model water demand and usage. The framework recognizes that a nexus of these elements is not only logical (given the inherent interdependencies) but also that reality dictates that a study of any practical good towards sustainability must consider their interaction. Ramos et al. [36] cataloged 23 recent studies (2012–2020) in which the CLEW framework was implemented in (i) planning water supply systems to meet future domestic, industrial, and agricultural demand, (ii) developing decarbonization pathways, (iii) investigating the impact of climate change on electricity generation, (iv) analyzing effects of environment flow (e-flow) directives on hydropower generation, and (v) investigating the effect of national energy policies on available water sources among others.

Several tools have been employed in literature to capture the various input elements of the CLEW framework. These include Soil and Water Assessment Tool (SWAT), Water Erosion Prediction Project (WEPP), Agricultural Policy/Environnemental eXtender (APEX), (Système Hydrologique Européen (MIKE-SHE), Hydrologic Modeling System (HEC-HMS), and Water Evaluation and Planning (WEAP). Recent studies investigating the effect of projected climate change on hydrological features have increasingly preferred the WEAP tool. For example, Asghar et al. [39], Opere [40], and Abera and Ayenew [41] investigated the impact of climate change on stream flow in the central Indus basins, Narok County, and Central Rift Valley basins, respectively, and they all obtained relatively high predictions of historical stream flows. Others, like Sridharan et al. [8,37,42,43] and Cervigni et al. [44,45], developed WBMs in WEAP to analyze the R. Nile flow under extreme climate change. These too achieved high agreement between modeled and historically observed flows and were able to implement several "*what if*" scenarios.

Therefore, this study developed a WBM in WEAP given its ability to: (i) model water supply at a catchment scale using different climatic, hydrological, and socioeconomic scenarios, (ii) integrate supply with different demand types, and (iii) implement ecological flow restriction regimes whilst catering for the various demand priorities and supply preferences.

2.3.1. WEAP Model Set-Up

The WEAP tool [46] was used to develop a WBM that accounts for the incoming, consumed, lost, and outgoing water within the catchments of the large HPPs. The model was built following a 5-part resilience evaluation framework, namely, threat characterization, component vulnerability analysis, system response, resilience evaluation, and evaluation of adaptation measures. The subsequent sections illustrate how these parts were modeled within WEAP, but first, this section details the general structure of the model.

The locations of HPPs were used to construct catchments in WEAP (as seen in Figure 8). The catchments were disaggregated into land classes and by selecting the rainfall–runoff

(simplified coefficient) method within WEAP, three key data types were required to characterize the catchments: crop coefficient, effective precipitation, and reference evapotranspiration ($ET_{ref}$). This study considered crop coefficient (0.87) and effective precipitation (79%) for all land types across all catchments as proposed by Sundin and Lindblad [9]. The precipitation was directly derived from ERA5 and CMIP6 data files as explained in Section 2.2.3 whereas the $ET_{ref}$ was computed using the FAO Penman–Monteith Equation (1) [47] and supporting Equations (2)–(9).

$$ET_{ref} = \frac{0.408\Delta(R_n - G) + \gamma\frac{900}{T+273}u_2(e_s - e_a)}{\Delta + \gamma(1 + 0.34u_2)} \times Days \tag{1}$$

$$T = \frac{T_{max} + T_{min}}{2} \tag{2}$$

$$e^o(T) = 0.6108exp\left[\frac{17.27T}{T + 237.3}\right] \tag{3}$$

$$e_s = \frac{e^o(T_{max}) + e^o(T_{min})}{2} \tag{4}$$

$$e_a = \frac{e^o(T) \times RH}{100} \tag{5}$$

$$\Delta = 4098\frac{e^o(T)}{(T + 237.3)^2} \tag{6}$$

$$\gamma = 6.65 \times 10^{-3}P \tag{7}$$

$$R_n = (1 - 0.23)R \tag{8}$$

$$u_2 = u_z\frac{4.87}{ln(67.8z - 5.42)} \tag{9}$$

where $Days$ are the days in a month, $R_n$ is net radiation at the crop surface [MJ/m$^2$/day$^-$] at 0.23 albedo, $R$ is net incoming radiation minus net outgoing [MJ/m$^2$/day$^-$], and $G$ is soil heat flux density [MJ/m$^2$/day], which is ignored in this study. $T$ is the mean daily air temperature at 2 m height [°C], $T_{max}$ is the maximum daily air temperature at 2 m height [°C], $T_{min}$ is the minimum daily air temperature at 2 m height [°C], $u_2$ is the wind speed at 2 m height [m/s], $u_z$ is the wind speed measured at $z$ (10 m) height [m/s], $e_s$ is the saturation vapor pressure [kPa], $e^o(T)$ is the saturation vapor pressure at the air temperature T [kPa], $e_a$ is the actual vapor pressure [kPa], $RH$ is the relative humidity (%), $\Delta$ is the slope vapor pressure curve [kPa/°C], $\gamma$ is the psychrometric constant [kPa/°C] and $P$ is the atmospheric pressure [kPa].

The model has 10 power plants and subsequently 10 catchments. Each of the catchments has an irrigation and municipal water demand totaling 20 demand sites. All power plants are modeled as run-of-river with each having a diversion from the R. Nile. The model incorporates another stream element, which represents the flow into L. Kyoga. L. Kyoga and L. Victoria were modeled as reservoirs with the latter located downstream of R. Nile's headrace and linked to the R. Nile through a transmission link and the former as an instream reservoir. The transmission link mimics the Nalubaale dam complex, which is operated by an Agreed Curve [13,22] represented in Equation (10). Within WEAP, L. Victoria level is a derived parameter, therefore this study used the lake level of the 'previous month' to restrict the flow of the 'current' month. The model has 42 other transmission links connecting the river to each demand site. Given that Kiira HPP has a small catchment

sandwiched between Nalubaale and Bujagali catchments, an extra transmission link was assigned to Kiira irrigation demand from the Nalubaale catchment.

$$Q = 66.3(H - 7.96)^{2.01} \tag{10}$$

where $Q$ is the outflow from Lake Victoria (m$^3$/s) and $H$ is the Lake level (m) for the 'previous' month.

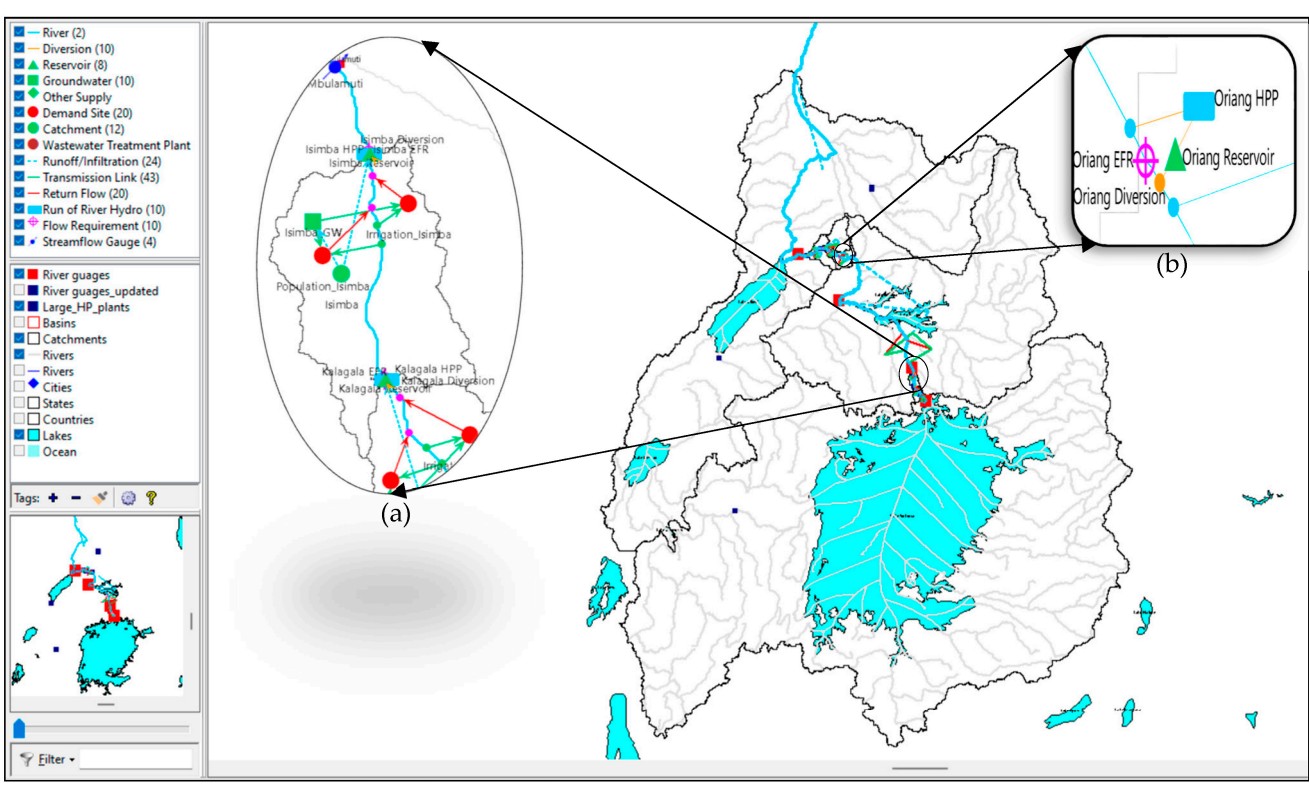

**Figure 8.** WEAP Model Showing R. Nile Basin with Insets of (**a**) Isimba Catchment Components and (**b**) a Typical arrangement of HPP Components.

Each of the catchments has a groundwater recharge station, which is supplied by a runoff/infiltration link. The model has 24 such links, with each catchment having at least two, one designated to run-offs at the catchment pour point and another to groundwater. Other runoff/infiltration links are designated to L. Victoria from the Nalubaale catchment, L. Kyoga from the Karuma catchment, the runoffs at the basin pour point, and another to a 'pseudo' catchment formed at the first gauging station near the R. Nile's headrace. The groundwater recharge was regarded to be 10% of the catchment runoff as indicated by others [8,9,48] with the rest designated to runoffs at the catchment's pour point. The exception to this allocation was made for the Nalubaale and Karuma catchments, which have large reservoirs (lakes) as explained in Section 2.3.3.

At all demand sites, return flow links were included with the unconsumed supply redirected back to the R. Nile. In addition, a flow requirement element was placed upstream of each diversion to a power plant. This was to ensure the provision of the minimum environmental flow requirements (EFRs) as required by the regulator. However, the study was not able to find data on the required EFR for individual plants. Therefore, for baseline assessment, the method proposed by Sridharan et al. [8] was adopted, which recommended an average flow requirement (AFR) derived from models by Smakhtin et al. [49], Tennant [50], Tessman [51] and two from Pastor et al. [52], as seen in Figure 9.

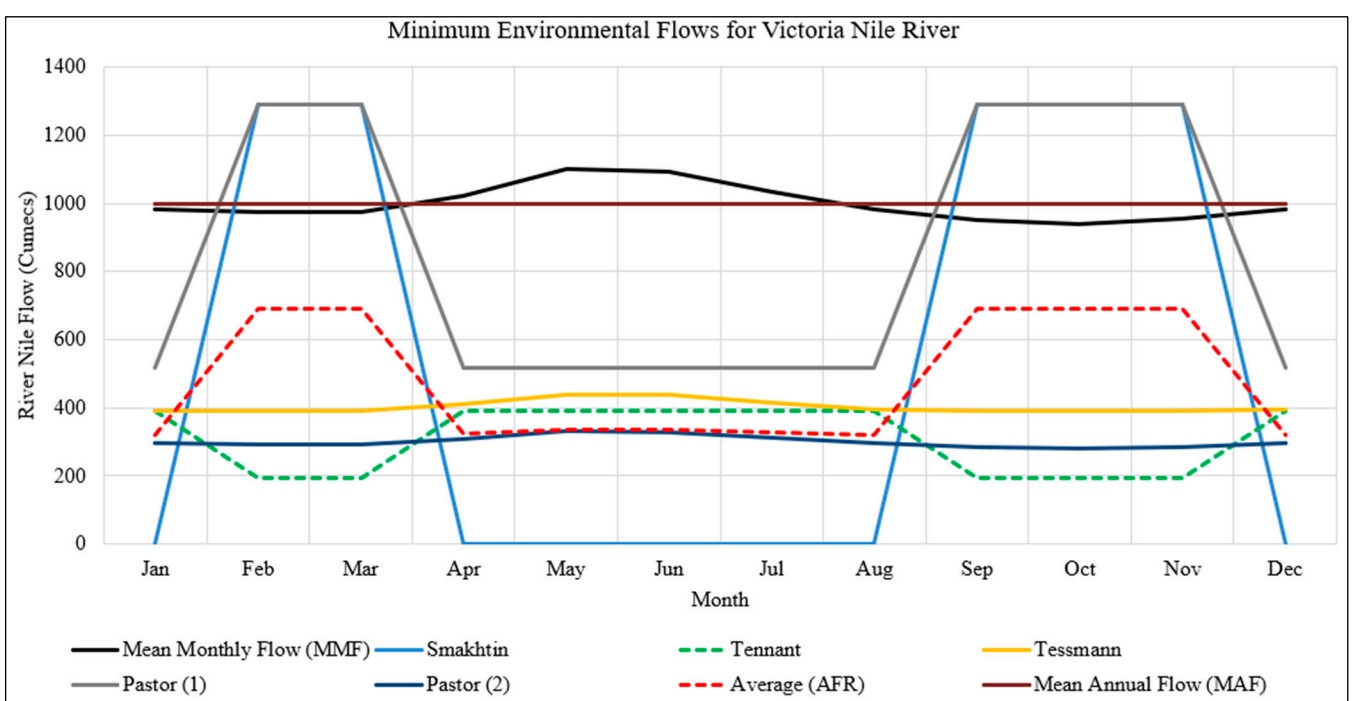

**Figure 9.** Minimum Environmental Flows Computed by Different Methods Using R. Nile Discharge at Jinja Pier.

In WEAP, water is allocated by three priorities. These are demand priority, supply preference, and distribution order. For demand priority, water allocated to competing demand sites, instream flow requirements, reservoirs, and hydropower generation can take up a priority of 1 to 99 with 1 signifying the highest and 99 the lowest priority. In essence, water will be allocated to a component with priority two if the demand for priority one is fully met. For components with the same priority, the available supply is divided equally. Supply preference criteria ensure that multiple supply sources for a given demand site are ranked. This means that a source with priority one will supply demand first and only be supplemented by the next priority source if it is not able to meet the demand. Lastly, the distribution order controls how water is internally distributed to branches within a demand site or catchment. In this study, only demand priority and supply preference criteria were considered.

Relative to all other demands, water for municipal consumption was considered priority one, priority two for irrigation and EFR, priority three for hydropower generation, and priority four for reservoir filling. For supply preference, irrigation demand was supplied first by groundwater and then by river abstraction. In contrast, municipal demand was supplied primarily by the river and then by groundwater sources. Altogether, 19 distinct '*what-if*' scenarios were created in the WEAP model. These represent the system under various conditions including the base year, reference scenario, extreme climate change, emission pathways, and adaptation, as explained in Section 2.3.2. In all scenarios, unless stated otherwise, the same assumptions were considered for consumption, growth rates, demand and supply priorities, crop coefficient, and effective precipitation.

### 2.3.2. Modeled Scenarios

All scenarios were considered within several categories namely, current, reference, extreme climate change, the impact of radiative forcing, and adaptation measures.

A. Current

The WEAP Model was constructed on a monthly resolution spanning the years 1981 to 2060. The current scenario, technically known as the "Current Accounts" in WEAP

software (version: 2021.0.1.10_), is illustrative of the supply and demand dynamics of the base year of the model. In this study, the Current Accounts characterized the CLEWs features as they were understood to be in 1981, and as such the year served as the starting simulation year for all other scenarios.

B.   Reference

The reference scenario represents the system evolution without any major intervention [19]. The study cycled climate data, from the ERA5 datasets, of the past (1981–2020) into the future (2021–2060). This scenario assumes that the future climate will be a replica of the past. The consideration of 40-year extents is informed by the recent IPCC report [35] that evaluated changes in climatic parameters by considering 20- to 40-year spans.

C.   Extreme Climate Change

Given that the main objective was to ascertain the resilience of hydropower plants under extreme climatic conditions, the study utilized the Climate Moisture Index (CMI) approach, which has been employed in several studies [8,53] to select GCMs that represent the wettest and driest future outcomes. CMI is a measure of aridity that combines the effect of rainfall and temperature projections [44]. The index ranges between −1 and +1 with the latter indicative of precipitation being higher than potential evapotranspiration (PET) rates and as a result, wet conditions are expected whereas the former indicates arid conditions. Alternative indexes, such as the Palmer Drought Severity Index and Surface Water Supply Index, for characterizing rainfall deficit have been described at length by Chong et al. [54].

This study used CMI values obtained for various GCMs within CMIP6 for years spanning 2021–2060. The climate data (temperature and precipitation) over the Nalubaale catchment were used given the massive influence that L. Victoria has on the R. Nile's observed flow within the basin. CMI values were computed as the percentage difference of PET from precipitation as seen in Equation (17). PET values for individual GCMs were obtained by using Thornthwaite Models proposed by [55,56] (see Equations (11)–(16)).

$$PET = PET_c \times f \tag{11}$$

$$PET_c = 16 \left[ \frac{10T}{I} \right]^a \tag{12}$$

$$I = \sum_{j=1}^{12} i_j \tag{13}$$

$$i = \left[ \frac{T}{5} \right]^{1.514} \tag{14}$$

$$a = 6.75 \times 10^{-7} I^3 - 7.71 \times 10^{-5} I^2 + 1.79 \times 10^{-2} I + 0.49 \tag{15}$$

$$f = \left[ \left( \frac{\theta}{30} \right) \left( \frac{h}{12} \right) \right] \tag{16}$$

$$CMI = \frac{P - PET}{P} \tag{17}$$

where $PET_c$ is the normative potential evapotranspiration values (mm/month), $PET$ is the adjusted potential evapotranspiration values (mm/month), $T$ is the monthly averaged temperatures (°C), $I$ is the annual heat index, $i$ is the monthly heat index, $a$ is the constant, $f$ is the adjustment factor, $\theta$ is the length of the month (days), $h$ is the duration of daylight (in hours) on the fifteenth of the month, and $P$ is the precipitation (mm/month).

For the Nalubaale catchment, CMIs for 86 SSP-GCM combinations were valid. Others were not considered, given that their CMI values were lower than −1. The authors suspect that in such GCMs, the temperature was considerably overestimated, or precipitation was overly underestimated. Figure 10 shows that ACCESS-CM2 from SSP5-8.5 represents the

probable driest conditions and INM-CM-8 from SSP3-7.0 represents the wettest conditions. Weather data from these two GCMs were used to contrast the production of hydropower plants at both extremes of the projected river discharge.

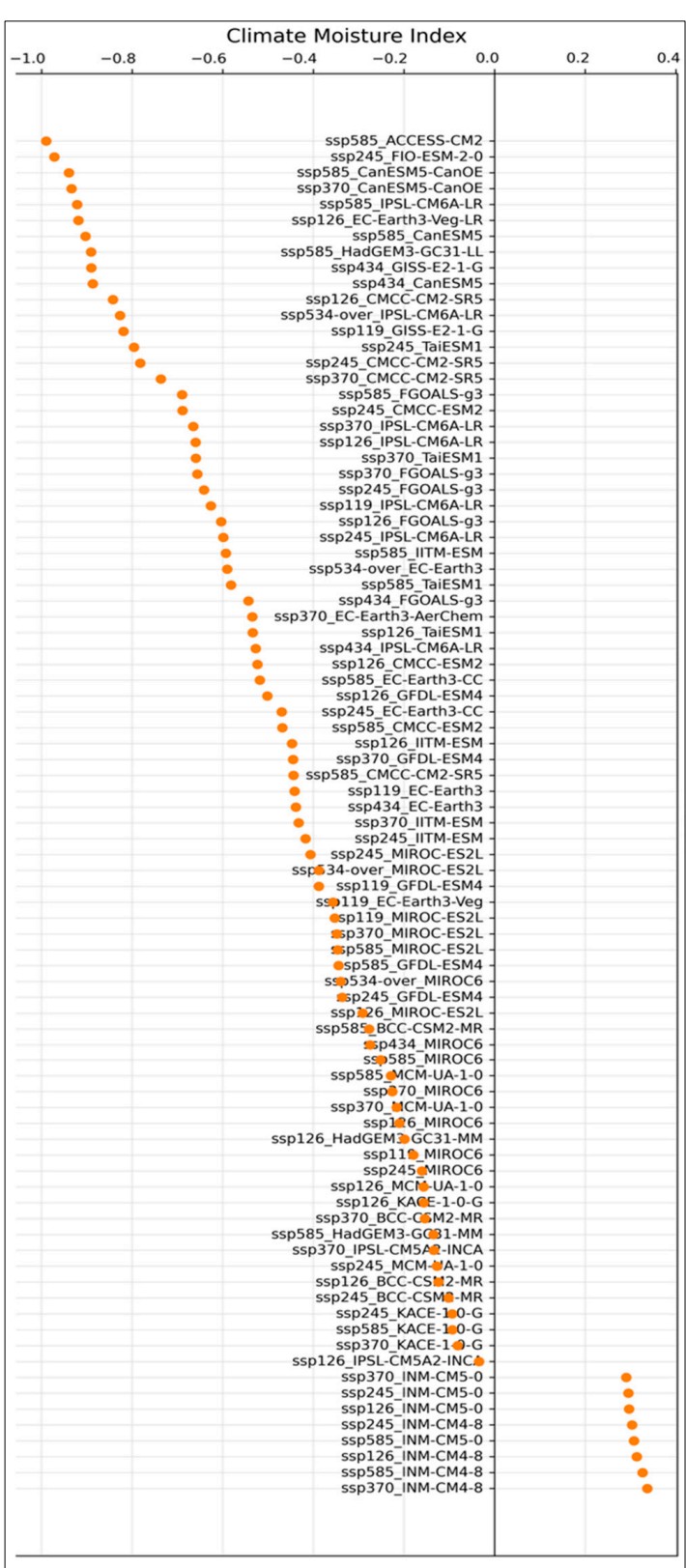

**Figure 10.** Climate Moisture Index for Various GCM-SSP Combinations.

A.    Impact of Radiative Forcing

The study ascertained how changes in radiative forcing could influence the expected plant generation as projected by the 'driest' and 'wettest' GCMs. Accordingly, 4 emission pathways (SSP1-2.6, SSP2-4.5, SSP3-7.0 and SSP5-8.5) were considered for both ACCESS-CM2 and INM-CM-8.

B.    Effectiveness of Adaptation Measures

The adaptation to extremely low river discharge was considered by three major interventions, namely, *robustness*, *responsiveness*, and *redundancy*. In each of these, three conditions were explored. For *robustness*, which is primarily intended to reduce exposure of the HPPs to extreme weather conditions, the study modeled this by adjusting the commercial operation date (COD) for planned power plants by 5 years ahead of schedule, and then by 5 and 10 years beyond schedule. *Responsiveness* was modeled by adjusting the EFR regime. The adjustments included the consideration of EFR at 50% and 75% of the monthly flows proposed in Section 2.3.1 as well as considering a constant EFR value of 150 m$^3$/s for all HPPs. As for *redundancy*, the volume of each reservoir for planned HPPs was scaled by 2, 5, and 10 times. The assessment of the effectiveness of these adaptative measures can be also interpreted as a sensitivity assessment of the model.

### 2.3.3. Model Calibration and Validation

The goal of calibration is to ensure that selected values of certain independent variables can reproduce the historical values of a dependent variable. Given that such a task and its output are inevitably characterized by uncertainties, statistical metrics are used to assess the accuracy of modeled outcomes. In WEAP, the calibration can be performed manually or by automation. However, given that the flow of the R. Nile from L. Victoria is controlled by a dam as explained in Section 2.3.1, the automatic tools could not be employed. Therefore, the calibration was performed by manually adjusting (trial and error) two key parameters: crop coefficient and groundwater recharge of the catchments with lakes (Karuma and Nalubaale).

Calibration was primarily performed on L. Victoria levels at Owen Falls Dam, which accounts for the R. Nile headrace outflow. The simulated levels were compared to the observed levels for the period 1981–2020. Following what was deemed a high goodness of fit, the model was validated by simulating river flows at the 4 flow gauging stations. To assess the goodness of fit between simulated and observed values, the Nash–Sutcliffe Efficiency ($NSE$) [57], coefficient of determination ($R^2$), normalized root means square error ($nRMSE$) and percentage bias ($PBIAS$) were employed. These are expressed as seen in Equations (18)–(21), and several studies [44,58,59] have provided some guidance on the interpretation of results as summed up in Table 11. In addition to the discharge at gauging stations, another form of validation involved hydropower generation for the period 2010–2020.

$$R^2 = \frac{\left( \sum \left[ X_i - \overline{X} \right] \left[ Y_i - \overline{Y} \right] \right)^2}{\sum \left( X_i - \overline{X} \right)^2 \sum \left( Y_i - \overline{Y} \right)^2} \tag{18}$$

$$NSE = 1 - \frac{\sum\limits_{i=1}^{n} \left( X_i - Y_i \right)^2}{\sum\limits_{i=1}^{n} \left( X_i - \overline{X} \right)^2} \tag{19}$$

$$nRMSE\ (\%) = \frac{100}{\overline{X}} \sqrt{\frac{\sum\limits_{i=1}^{n} \left( X_i - Y_i \right)^2}{n}} \tag{20}$$

$$PBIAS(\%) = 100 \left( \frac{\overline{Y} - \overline{X}}{\overline{X}} \right) \tag{21}$$

where $X_i$ and $Y_i$ are the $i^{\text{th}}$ observed and simulated monthly discharge data, respectively; $\overline{X}$ and $\overline{Y}$ are the mean of the observed and simulated monthly discharge data, respectively, and *n* is the total number of observations.

**Table 11.** Interpretation of Goodness-of-fit Values of Statical Indices.

| Performance Rating | PBIAS (%) | nRMSE (%) | $R^2$ | NSE |
|---|---|---|---|---|
| Unsatisfactory | $\|PBIAS\| > 25$ | $nRMSE > 80$ | $0 \leq R^2 \leq 0.2$ | $0 \leq NSE \leq 0.2$ |
| Satisfactory | $15 < \|PBIAS\| \leq 25$ | $60 < nRMSE \leq 80$ | $0.2 < R^2 \leq 0.6$ | $0.2 < NSE \leq 0.6$ |
| Good | $5 < \|PBIAS\| \leq 15$ | $40 < nRMSE \leq 60$ | $0.6 < R^2 \leq 0.75$ | $0.6 < NSE \leq 0.75$ |
| Very Good | $\|PBIAS\| \leq 5$ | $nRMSE \leq 40$ | $0.75 < R^2 \leq 1$ | $0.75 < NSE \leq 1$ |

Source of data: [41,44,58–60].

The crop coefficient was manually adjusted between 0 and 1.3 and it was found that the best correlation between simulated and observed flow was achieved at 0.87 as previously documented by [9]. Secondly, it was observed that a large portion of simulated flow within catchments with large reservoirs (Nalubaale and Karuma) was registered at the catchment outflow points. This meant that the two lakes were always at their maximum levels. Therefore, within the Nalubaale catchment, a proportion of the runoff was directed towards groundwater recharge using proposed models in Equations (22) and (23). The rationale of the models for the period 1981–2020 (past) was to make simulated lake level closely similar to historical observations, whereas from 2021 to 2060 (future), the models were cycled not to reintroduce new uncertainties in the study. Although this concept is novel by the proposed models and redirection of flow into a groundwater node for reservoir catchments, it builds on the work of others [13,21–23] who observed that the flow of the R. Nile can only be calibrated by directly controlling the volume and levels of L. Victoria and L. Kyoga. In addition, Hughes et al. [61] inferred that a WBM could be calibrated "with either interflows or groundwater outflow as the dominant process". Similarly, Dehghanipour et al. [62] calibrated their WBM by indirectly controlling groundwater storage using estimated pumping requirements.

$$Loss~to~Groundwater = \begin{cases} 0.57P_T, & 1981\text{–}1984,~2021\text{–}2025 \\ 0.64P_T, & 1985\text{–}1989,~2026\text{–}2030 \\ 0.57P_T, & 1990\text{–}1994,~2031\text{–}2035 \\ 0.40P_T, & 1995\text{–}1999,~2036\text{–}2040 \\ 0.42P_T, & 2000\text{–}2004,~2041\text{–}2045 \\ 0.40P_T, & 2005\text{–}2009,~2046\text{–}2050 \\ 0.40P_T, & 2010\text{–}2014,~2051\text{–}2055 \\ 0.30P_T, & 2015\text{–}2020,~2056\text{–}2060 \end{cases} \tag{22}$$

$$P_T = P \times A \tag{23}$$

where $P_T$ is the total rainfall over the catchment area (MCM/month) and *A* is the catchment area (m$^2$).

The model represented in Equations (22) and (23) are not a representation of the groundwater recharge mechanism within the lake but simply a tool implemented to calibrate lake levels at the R. Nile headrace. For the Karuma catchment, the runoff fraction directed towards groundwater nodes was set to 99%. This was because, within WEAP, L. Kyoga operated at full capacity, causing the overflow, runoffs, and inflows to be part of the river discharge with no storage at all. A previous study [9] dealt with this problem by overestimating the volume of the lake to curtail it from overflowing. The method used in this study achieves similar results without changing the physical orientation of the lake, which governs the evaporation process. Nonetheless, the authors, recognize this as the most pragmatic (aim-driven), and perhaps weakest, part of the modeling exercise, which could be moderated by using downscaled and bias-corrected climate model data, or by em-

ploying observational data for lakes' precipitation, evaporation, and inflow. Alternatively, the calibration process proposed by Vanderkelen et al. [21,22] can be adopted, which could lead to a closure of the WBM without employing groundwater affixation; however, even so, one would need to restrict (fix) the operational range of the lake levels or volumes. In effect, whichever calibration method is adopted, the levels (and by extension, the volumes) of the lakes must be controlled. Once the WBM is calibrated and the historical flows reproduced with a good level of certainty, the projected flows are solely determined by input GCM data.

## 3. Results and Discussion

This section presents and discusses the findings and results of projected the R. Nile flow and subsequent hydropower generation for the modeled scenarios. Section 3.1 describes the calibration of the model and its validation under the cycled climate data scenario and Section 3.2 presents the river flow and the associated electricity production under the wettest and driest conditions. Section 3.3 discusses how different emission pathways affect river flows projected from extreme future conditions. Finally, Section 3.4 presents the effect of adaptive measures on the scenario with the lowest cumulative projected flows.

### 3.1. Calibration and Validation

In the past, different approaches were employed to calibrate and validate WEAP models. For example, Fernández-Alberti et al. [63] used data from four rain gauges; one-half of the dataset (2000–2007) was used to calibrate the model and the other half (2008–2015) for validation. Sridharan et al. [8,64] used the entire dataset to calibrate the streamflow but provided no validation results in their studies. Contrary to both approaches, this study used three data types: (i) lake levels for calibration, (ii) river flow, and (iii) hydropower generation for validation.

Table 12 and Figure 11 show the results from the calibration and validation exercise. A high degree of comparability ($NSE = 0.75$, $R^2 = 0.8$, $PBIAS = 0.4\%$, $nRMSE = 2\%$) was observed between simulated and observed monthly lake levels. Both $NSE$ and $R^2$ indicate that the simulated levels are highly correlated and are nearly similar to observed levels, whereas $PBIAS$ and $nRMSE$ indicate that the residuals between both sets of values are relatively small. It was on this basis that the flow at four gauging stations was validated. The simulated flow at the headrace was found to be highly comparable to the observed flow ($NSE = 0.77$, $R^2 = 0.8$, $PBIAS = -3.9\%$, $nRMSE = 11\%$). It should be noted that the color coding of results in Table 12 is in reference to the performance ratings in Table 11.

**Table 12.** Statistical Performance of Measured and Modelled River Nile Discharge, Lake Victoria Level, and Hydropower Generation from the Large Plants.

| Phase | Stations | *NSE* | $R^2$ | *PBIAS* (%) | *nRMSE* (%) |
|---|---|---|---|---|---|
| Calibration | L. Victoria Level (1981–2020) | 0.75 | 0.8 | 0.4 | 2 |
| Validation | 1. River Discharge (1981–2020) | | | | |
| | (i) Jinja Pier | 0.77 | 0.8 | −3.9 | 11 |
| | (ii) Mbulamuti | 0.03 | 0.3 | −5.6 | 24 |
| | (iii) Masindi Port | −0.73 | 0.1 | 6.6 | 26 |
| | (iv) Paara | 0.23 | 0.3 | 2.3 | 31 |
| | 2. Hydropower Generation (2011–2020) | | | | |
| | (i) Variable EFR | −0.26 | 0.8 | 22.8 | 32 |
| | (ii)] Constant EFR | 0.4 | 0.9 | −12.8 | 22 |

In previous related studies, Sundin and Lindblad [9] did not present any method or results for calibration and validation of the R. Nile discharge or L. Victoria levels. In Sridharan et al. [8], calibration results were presented and they cited a method from Li et al. [65]. However, there is nothing in that study [65] to suggest that the flow of L. Victoria could be calibrated. It is not well understood how Sridharan et al. [8] calibrated the flow

from L. Victoria without employing a dam control model. The significance of calibrating L. Victoria levels is evident in Table 12 and Figure 11, which show that a high goodness of fit between observed and simulated lake levels translates into high comparability of simulated lake outflows.

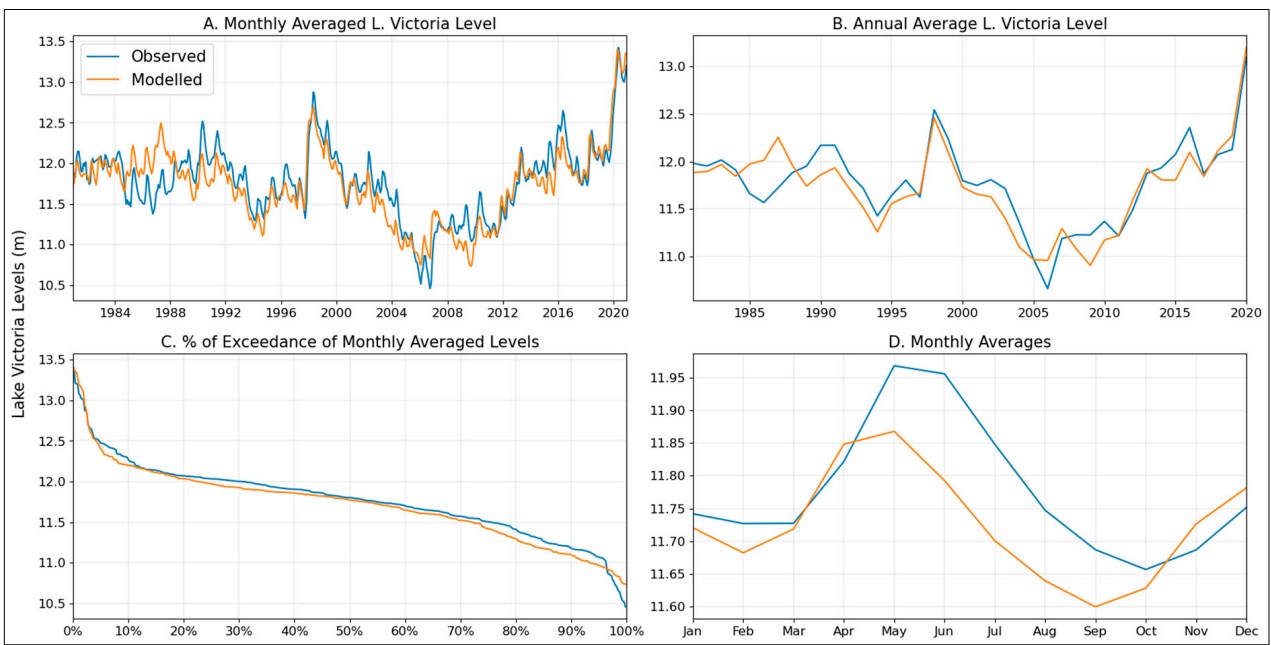

**Figure 11.** L. Victoria Simulated and Observed Levels at Owen Falls Dam for Period 1981–2020.

As seen in Table 12 and Figure 12, for gauging stations further downstream of the lake outflow, the correlation between simulated and observed flows was relatively low, although the residuals were within the acceptable range. The general trend indicates that the further away from the headrace, the higher the variance. Three reasons may be attributed to these observations. First, at the headrace, the flow is mainly governed by dam operations unlike downstream where consumption patterns and runoffs from adjacent catchments play significant roles in the observed stream flow. Second, the study used reanalysis data without downscaling or bias correcting, given that such methods rely on observed weather data, which were unavailable at the time of the study. Downstream of the lake, the simulated catchment runoffs are highly sensitive to input weather data, and the larger the biases within the data, the larger the deviation of simulated from observed discharge. In the past, several studies [21,22] observed the inability of unprocessed reanalysis data to reproduce observed flows within the basin. Third, the WEAP model is rigid in its operation of reservoirs. Water can only be drawn out of lakes to meet a particular demand or if the lake overflows. In addition, the model does not consider interyear storage such that a deficit in a dry year can be met by surplus from pervious wet years. This meant that the simulated flows largely depicted downstream demand more than the natural flow of the river.

The WEAP model was also validated by quantifying the generation from large hydropower plants. Two scenarios were used in this exercise; the reference scenario, which employed a variable EFR as explained in Section 2.3.1 and a constant EFR (150 m$^3$/s) as is likely to be in practice. Figure 13 and Table 12 show that generation from plants was better predicted by a constant EFR ($NSE = 0.4$, $R^2 = 0.9$, $PBIAS = -13\%$, $nRMSE = 22\%$) than the variable EFR ($NSE = -0.3$, $R^2 = 0.8$, $PBIAS = 23\%$, $nRMSE = 32\%$). Nonetheless, the variable EFR was adopted and applied to other scenarios, since it led to better prediction of historical (1981–2020) river discharge.

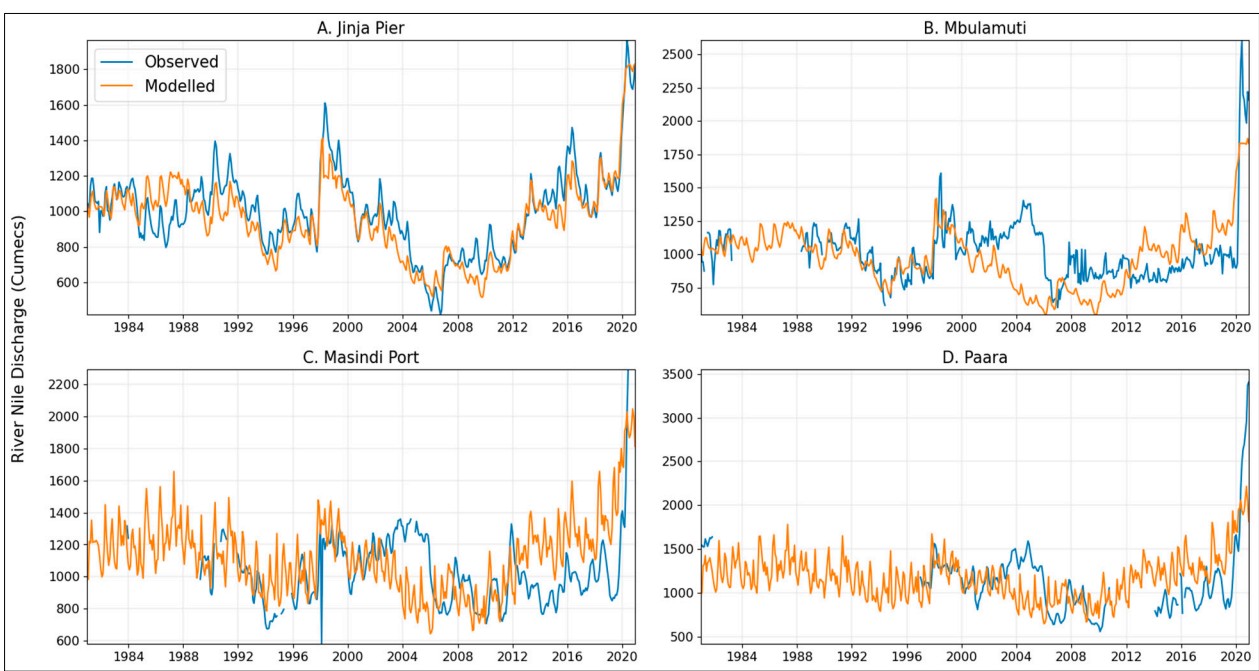

**Figure 12.** Simulated and Observed River Nile Flow at Various Gauging Stations.

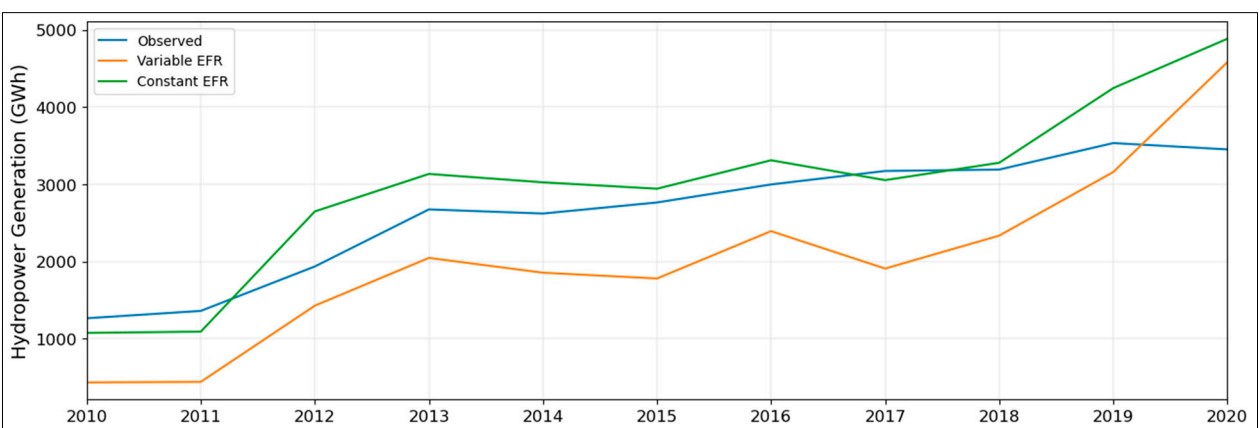

**Figure 13.** Modelled and Observed Hydropower Generation under Two EFR Conditions.

Note that $NSE = 1$, suggesting that modeled parameters are equivalent to observed values such that the variance is zero. $NSE = 0$ indicates that the modeled values are as good as the observed mean, whereas an $NSE < 0$ shows that the mean predicts better values than the model [44]. In this respect, the prediction of the flow at Masindi Port and HPP generation (under variable EFR) performed less than if the mean of the observed values were assumed across the modeled period. Nevertheless, given that the $RMSE$ of all validated parameters remained <40% infers that the model has its usefulness for long-term prediction and planning even at those stations where the goodness of fit was weak.

In essence, the model performed exceptionally well in predicting the flow at the river headrace. The residuals within the simulated lake levels, river flow at all gauging stations, and HPP generation are, at the least, within what is considered satisfactory. However, simulated flow at Mbulamuti, Masindi Port, and Paara exhibit low correlation and considerable variance, in part due to biases within the data, the inadequacy of applied assumptions to reflect reality, and the inherent flexibility limitation of the WEAP tool. Moreover, the biases seem to be most significant and sustained in the period 2003–2006. This observation was also reported by JICA [13], in which they attributed the deviation to

poor calibration of lake levels and outflows. The JICA study also reported that in 2007, the agreed curve had been adjusted to deal with flooding downstream of the dam. Therefore, a significant portion of biases observed in this study is a direct result of changes made to the Owen Falls dam operations in the past. Unfortunately, some of those changes could not be outrightly depicted.

### 3.2. Hydropower Generation under Projected Extreme Climate

This section achieves four aims: (i) it juxtaposes the two projected R. Nile flows under extreme climatic conditions with a reference scenario; (ii) it quantifies the projected changes in the flow thereof; (iii) it discusses the likelihood of occurrence of the projected extremes; and (iv) it compares flows of the extreme scenarios to short-term observed 'future' flows. As described in Section 2.3.2, the CMI values were used to select climate models, which could predict the likely wettest and driest conditions. The GCM models ACCESS-CM2 from SSP5-8.5 and INM-CM-8 from SSP3-7.0 were found to represent the projected driest and wettest conditions, respectively.

The CMI approach achieved the objective of modeling the flow of the river at extreme climatic conditions necessary for investigating the resilience of HPPs and the effectiveness of adaptation measures. However, it also brought to the fore some concerns regarding the validity of CMIP6 data. First, it can be seen in Figure 14 that from the year 2021, the flow for the two extreme conditions widely varies from each other and the reference case. That is, if future climatic conditions are to follow the projection of SSP3-7.0 INM-CM4-8 (hereafter also referred to as "B3"), the river discharge will grow exponentially until 2050, and thereafter a cyclical trend will be observed. On the other hand, if future climatic conditions follow the trend projected by SSP5-8.5 ACCESS-CM2 (hereafter also referred to as "A4"), the river discharge will drastically plummet between 2021 and 2035 and thereafter an increase will be observed, although the discharge would remain significantly lower than for the reference case.

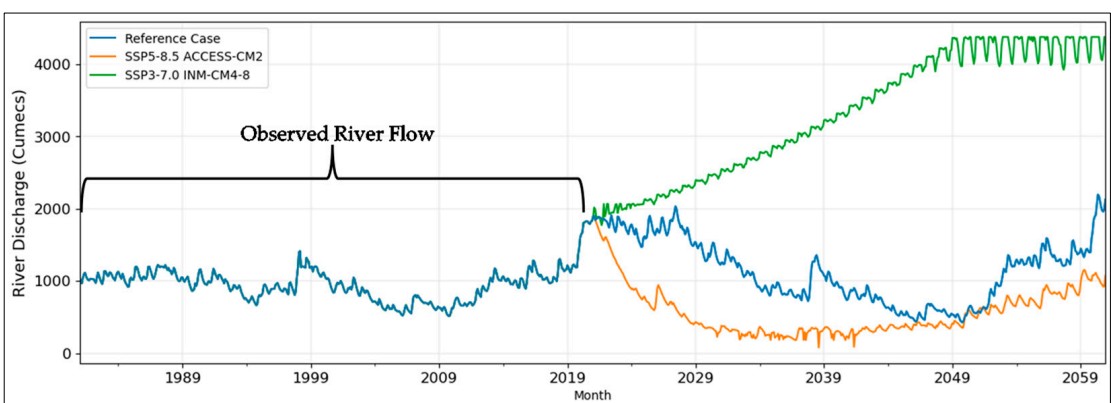

**Figure 14.** Monthly Average Discharge at L. Victoria Outflow for Reference, Driest and Wettest Scenarios.

Table 13 shows the disparity of the extreme scenarios relative to the reference case. The minus "−" sign denotes a decrease and the plus "+" sign, an increase. In the future (2021–2060), A4 predicts a decrease in the mean, minimum, maximum, 50th percentile (P50), and 10th percentile (P10) flows relative to the reference case (50%, 82%, 14%, 65% and 55%), respectively. In contrast, B3 predicts a future with extremely large flows with the mean flow of the river rising by 188% compared to the reference case. Note that the P50 and P10 flows are included in this analysis because they serve as proxies for HPP design flow and EFR, respectively. Similar results have been reported by Vanderkelen et al. [21], who modeled daily outflows using GCMs from RCPs 4.5 and 8.5 and found them to range between −85% to 229% of the historically observed flows. In addition, the phenomenon

of the river flow falling between 2021 and 2050, as seen in Figure 14 (reference case), and thereafter rising is comparable with findings of other studies [8,21,66,67].

**Table 13.** Projected R. Nile Flow at the Headrace Under Extreme Climatic Conditions Relative to Reference Case (2021–2060).

| GCM/Stats | Mean | Min | Max | P50 | P10 |
|---|---|---|---|---|---|
| A4 (Driest) | −50% | −82% | −14% | −65% | −55% |
| B3 (Wettest) | +188% | +313% | +99% | +186% | +269% |
| Historical Reference case (1981–2020) | +20% | −17% | +20% | +20% | −15% |

In addition, in comparing the historical reference case (1981–2020) with the future reference case (2021–2060), the future mean, maximum and P50 flows are all projected to increase by 20% and, in contrast, a decrease in the minimum and P10 flows are projected to be 17% and 15%, respectively, as seen in Table 13. In part, the projected increase is attributed to the river flow being strongly dependent on lake levels. Leading up to 2020, the flow of the river exponentially increased following a drastic increase in the lake levels but the relatively normal cycled climate for the period 2020–2038 meant that the lake levels of the immediate 'future' were projected to remain relatively high and, as such, facilitated the high river flows. Second, the simulation of L. Victoria outflow within WEAP is informed by demand, part of the increment in river outflow indicated in Table 13 is an indication of growth in water consumption downstream of the headrace.

Regarding the likelihood that either one of the extreme climate scenarios would materialise and indeed lead to results indicated in this study, it is hard to tell for the long term but it is unlikely that the driest scenario would be as drastic in the short to medium term as observed in Figure 14. Based on the three Mann–Kendal Tests (original, seasonal, and multivariate tests), which are used to test for trends within time series data, there is no detection of any trend at 0.05 significance in the historical headrace flow spanning the period 1948–2020. This infers that the river flows for the last 70 years have been relatively stable. Moreover, moving averages of one and five years of observed river discharge (seen in Figure 15A,B) show that in the 1960s, the river had a drastic increment in its flows, which was followed by a 50-year downward trend, although every 5 years (except for 2000–2010) there was an observed spike. From 2007 onwards, the river flow increased relatively faster than it had previously decreased. The sharp rise in the 1960s was attributed to excessive rainfall and increased tributary inflows [24], although it has been suggested from another study [68] that part of the rise could have resulted from computation errors given that the lake levels rose beyond the dam gauge limit at the time. Other than the period 2000–2006, there is no precedence for the drastic plummeting of river flows, and even so, the flows did not remain low for long.

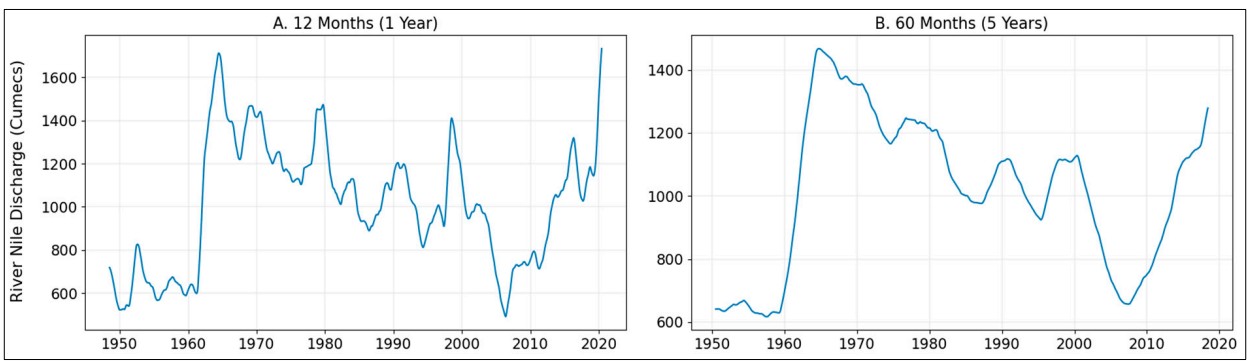

**Figure 15.** Moving Average of Monthly Flows for R. Nile at L. Victoria Outflow for Period 1948–2020.

This analysis is further complicated by the fact that when anecdotal observed data for 17 months spanning January 2021 to May 2022 are compared to both extremes and reference scenarios, the observed river flows seem to be better correlated to the simulated flows of the driest scenario as seen in Figure 16. Moreover, the mean flow for the driest condition (2021–2022) is 72% higher than the observed flow within the historical reference scenario (1981–2020). Given that only 17 months of observed data are available, there is not so much that the authors can deduce on the future river flow trend leading to 2060. It does, though, appear that the river has a 50-year cycle in which flows rise and fall given that the rise in the 2010s seems similar to what happened in the 1960s. However, this observation is inconclusive since it is only based on a single cycle, and consideration of other influential parameters, such as dating river sediments, was beyond the scope of this study.

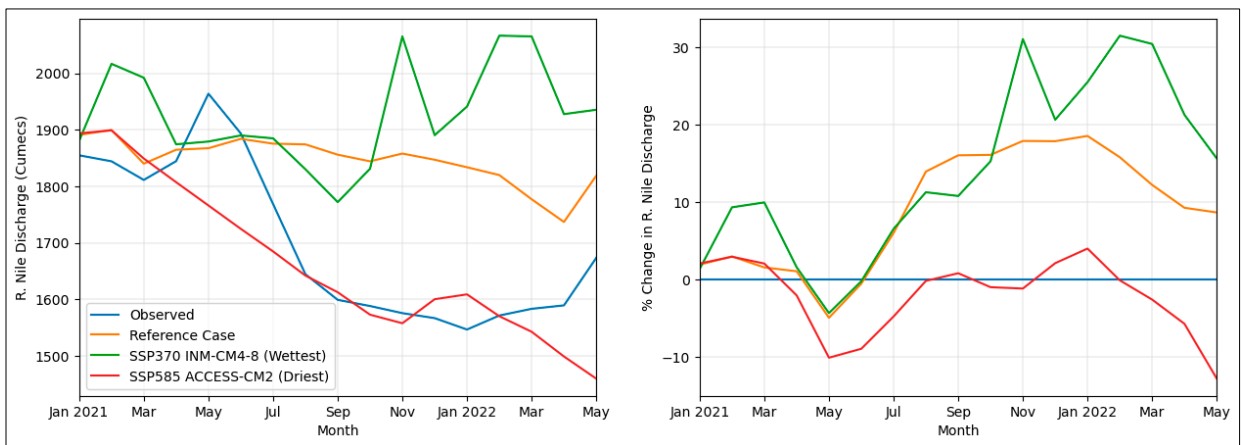

**Figure 16.** Comparison of R. Nile Observed Flow at L. Victoria Outflow for the period 2021–2022 with Selected Simulation Scenarios.

The GCMs B3 and A4 emerge from SSP3 and SSP5, respectively, which are well outside the preferred SSP1 (colloquially known as the "green road") or SSP2 (known as the "middle-of-the-road") scenarios. Essentially, GCMs developed under SSP3 and SSP5 take into consideration that progressively the objective of climate change mitigation will take a back seat as countries prioritise energy security and local development commitments ahead of global climate goals. It is, therefore, paradoxical that this study finds that GCMs developed on such common assumptions provide widely varying results. It might be that B3 and A4 are not well adapted for the study area and, in that respect, the validity of models within CMIP6 to represent global climate as remarked in several studies [35,69] was overstated. If indeed so, this detracts from the study from primarily being one of investigating HPPs' operations under extreme climate to one of investigating HPPs' operations under projections of extremely biased GCMs. The latter, although intellectually stimulating, might not be of practical benefit if the GCMs employed in the study are not able to predict future climate with a good level of certainty.

Although the results presented in this section are mixed and engender several questions all of which are not resolved, the study is useful in demonstrating the projected extreme flows reflective of the recent state-of-the-art data, tools, and methods. It then begs the question of how the simulated flows are affected by different emission pathways.

### 3.3. Impact of Projected Climate Change on Hydropower Generation

The results presented in this section show the impact of emission scenarios on projected river flows and subsequent hydropower production. GCMs from the family of the two extreme scenarios as discussed in previous sections are used such that A1, A2, A3, and A4 represent, in increasing order of radiative forcing, ACCESS-CM2 (dry scenario) from SSP1-2.6, SSP2-4.5, SSP3-7.0, and SSP5-8.5, respectively, as is similarly B1, B2, B3, and B4 for INM-CM4-8 (wet scenario).

The study shows that generally, the flow of the river and the respective generation from hydropower will increase with increasing radiative forcing. This is most pronounced with GCMs that predict a dry scenario (ACCESS-CM2) as demonstrated in Figure 17A–C. Except for A3, the flow of the river is projected to increase as demonstrated by its mean, P50, P10, and cumulative flow. The effect of radiative forcing for the wet scenario (INM-CM4-8) is such that the increase is most evident in the simulated minimum flows (Figure 17D–F). Subsequently, the wet scenario is projected to lead to an increase in hydropower generation as opposed to the dry scenario, which predicts an immense decrease. Even so, under dry conditions, the general effect of forcing is such that the higher the emissions, the more the production for hydropower plants, as seen in Figure 18A,B. This increase in precipitation with the projected increase in temperature, contrary to expectations, has previously been referred to as the "East African climate paradox" [21].

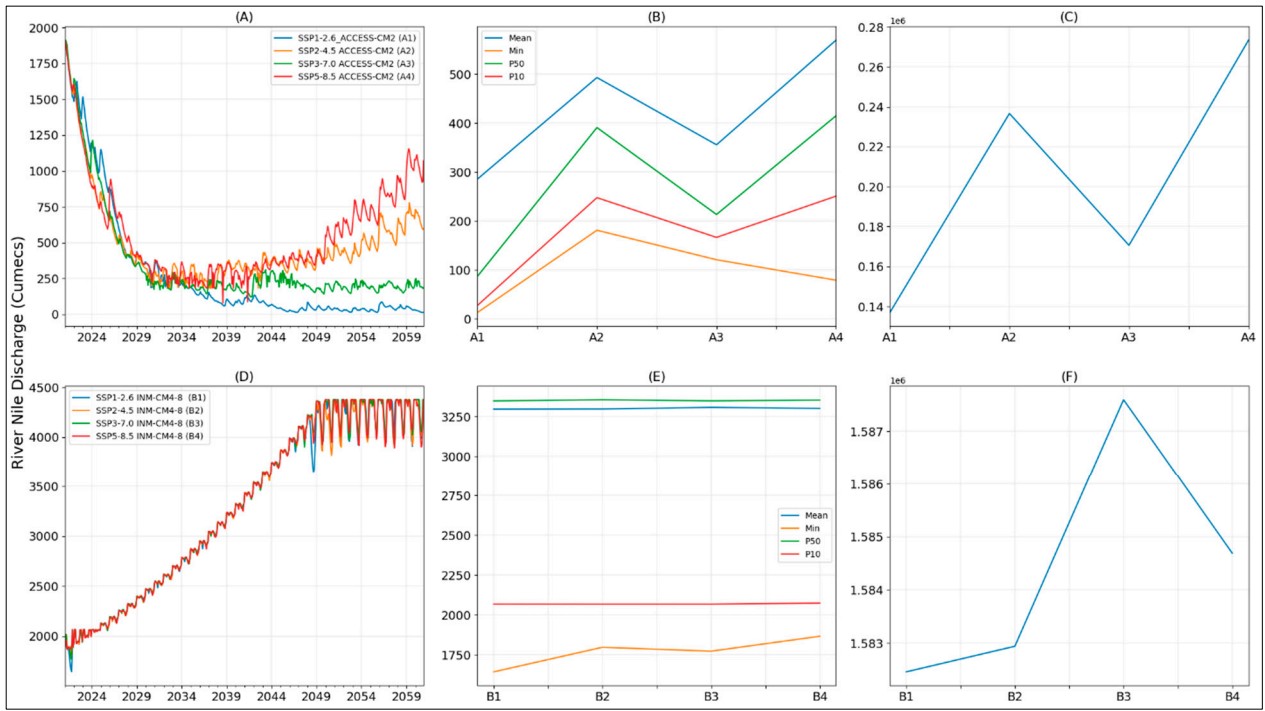

**Figure 17.** Comparison of Simulated Flow of R. Nile at L. Victoria Outflow Under Varying Radiative Forcing Scenarios for Period 2021–2060. (**A,D**) represent Monthly Flows Under Extreme Dry and Wet Conditions Respectively. (**B,E**) show the Trend of the Mean, Minimum, 50th, and 10th Percentile Flows of the Extreme Dry and Wet Conditions Respectively. (**C,F**) show the Cumulative Average Monthly Discharge for Dry and Wet Conditions.

This study estimates that under dry conditions with varying emission pathways, hydropower generation will reduce by 58 to 91% over the next 40 years, whereas under wet conditions, the increase could be about 53%, as seen in Figure 18. These results are comparable to Sewagudde [67], who projected a 50% fall in the net basin supply for the L. Victoria outflow by 2100. In addition, several studies [21,70] report a generally positive impact of increasing radiative forcing on L. Victoria outflows and a subsequent increase in generation. In contrast, other studies have reported slight variations between the driest and wettest outflows and generation. For example, Tate et at., [66] projected that outflow at the two extremes could be between −2.9% and 6.3%, comparable to Sridharan et al. [8], who projected hydropower generation to only fall by 2.6% under the driest scenario and increase by 11.6% under the wettest. Nonetheless, the underlying observation in most studies [66,70,71] estimating flows in the Nile basin is that with projected climate change, the flow will increase but so will its variability.

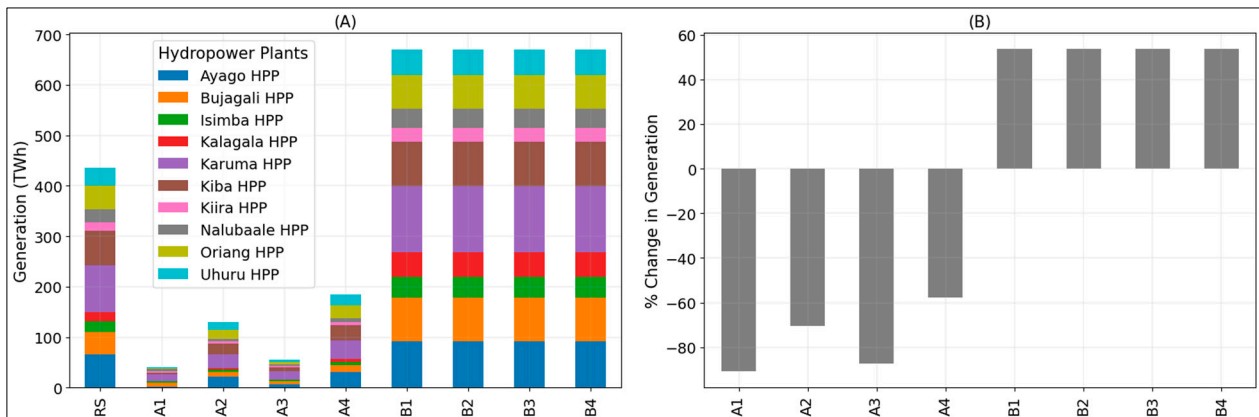

**Figure 18.** Simulated Hydropower Plants Generation for Period 2021–2060: (**A**) under Reference Scenario (RS), Dry (A1, A2, A3, and A4) and Wet Scenarios (B1, B2, B3, and B4), and (**B**) the % Change in Generation of the Dry and Wet Scenarios Compared to the Reference Scenarios.

### 3.4. Impact of Adaptation Measures on Hydropower Generation

Until Section 3.3, SSP5-8.5 ACCESS-CM2 (A4) was regarded as the 'driest' scenario. This was based on it having the lowest CMI value. However, due to intermodel variability from various emission pathways, it can be observed in Figures 17 and 18 that SSP1-2.6 ACCESS-CM2 (A1) was found to have cumulatively the lowest flows and subsequently the lowest projected hydropower generation. It is on this basis that adaptation measures were applied on A1 as duly explained in Section 2.3.2 (E). The general observation, as seen in Figure 19, is that *responsiveness* had a positive significance on generation, no effect from *redundancy*, and mixed results from *robustness*.

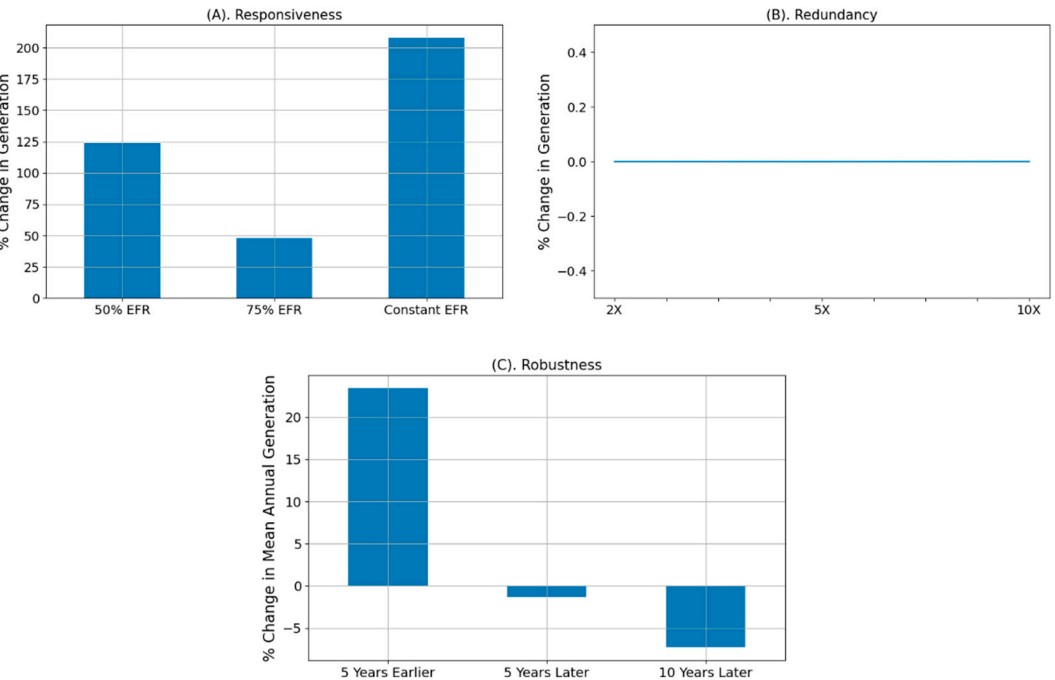

**Figure 19.** Effects of Adaptation Measures on A1 Scenario. (**A**) Is the Responsiveness Measure, which Shows the Change in the Cumulative Generation (2021–2060) when Environment Flow Rates are Varied. (**B**) Demonstrates the Effect of Increased Redundancy in the System by Expanding Reservoir Volumes of Planned Hydropower Plants by 2, 5, and 10 times. (**C**) Shows Results for the Robust Case by Changing the Commercial Operation Dates of Planned Power Plants.

Three *responsive* measures were explored by controlling the EFR at all power plants. The scenario of having a constant EFR of 150 m$^3$/s as seen in Figure 19A, which is 16% of the mean of historically observed flows, provided the largest increase (207%) in HPP generation. Applying 50% EFR as used in modeling baseline conditions to A1 resulted in a 124% increase and 75% of EFR resulted in a 48% increment of the cumulative generation across the period 2021–2060. Concerning *redundancy*, a change of reservoir volume for planned power plants to 2, 5, and 10 times, demonstrated no significant effect on future generation (Figure 19B). For *robustness*, the study demonstrated that if the commercial operation dates (CODs) of planned power plants were to be moved to 5 years earlier than scheduled, a 23% gain on annual mean generation per power plant would be realized. On the contrary, moving the CODs 5 and 10 years beyond the operational schedule would lead to respective losses of 1.3% and 7% of the normalized average annual plant generation (Figure 19C).

The *responsive* measures increased generation by directing a portion of the water away from ecological purposes to electricity production. In real life, this would be quite an unpopular and perhaps imprudent decision if such measures were to be taken for a long time and if the effects on river biodiversity were to be severe. The consideration of such measures would also depend on the distance between the intake and tailrace. For power plants like Nalubaale, Kiira, Bujagali, and Isimba, the intake and tailrace are only separated by a few meters, in contrast to 8 km for Karuma. In the instance of the former, deviating a portion of water meant for EFR into the generation facilities could do little harm to nature than in the latter's case where not only will the river biodiversity be gravely affected but so would thousands of people who depend on the river for their livelihood. In the context of this study, the 150 m$^3$/s seems sufficient for the textbook EFR [50] of the R. Nile in Uganda; however, without knowing what is mandated by the regulator or that which is sufficient for river ecology, this EFR serves at best as an indication of the sensitivity of flow on hydropower production during dry conditions.

The study observed that considering cases for larger reservoir volumes for planned power plants did not in any way affect generation during A1 conditions. This is because only two power plants were considered in this scenario (Kalagala and Uhuru) with a combined reservoir storage volume of 48 MCM. If both plants are operational at the same time and all storage is designated to generation, then the reservoir would only support 6 h of hydropower generation under A1 climate conditions, but it would require an average of 47 h to fill up the storage again. These values drastically increase as the volume is expanded such that at 10 times the planned storage, the plants would generate 61 h compared to the 468 h required for a refill. In short, the model does not compute any gains in expanding reservoir volume given that under the A1 scenario, reservoirs never fill up. There is never enough water to be designated for other uses and also for storage. That said, in reality, an expansion of reservoir volume is bound to help increase the generation especially if the flow variability increases. However, these gains could not be quantified in this study since the model was developed on a monthly temporal scale, which encompasses both phenomena of reservoir-aided generation and time required for filling.

If the future is as dry as simulated by the A1 climate scenario, the results seem to suggest that the robust adaptation measure would be to construct Ayago, Kiba, Oriang, Kalagala, and Uhuru five years ahead of their schedule. This means that the scheduled plants would all be constructed by 2034, which falls within the period of relatively high flows, as illustrated in Figure 17A. This could particularly be of interest to developers given that the power plants would stand to generate a substantial amount of their revenues during the early years (2021–2040). However, given that the financing of large hydropower plants is usually locked up in long-term contracts (≈50 years), a guarantee of good revenues of ≈20 years, as predicted by this study, might not be considered a huge incentive.

*3.5. Conclusions*

This study aimed at assessing Uganda's hydropower system's resilience in light of projected climate change and the effectiveness of adaptation measures for extreme aridity. A coupled water-balance and hydropower model was developed in the WEAP tool to model four major long-term scenarios: (i) cycling historical weather to the future, (ii) extreme dry conditions, (iii) effects of emissions pathways, and (iv) effectiveness of adaptation. The study focused on the large power plants constructed and planned along the R. Nile.

It was demonstrated that river flows for the last 70 years have been relatively stable and there is no apparent historical trend between climate change and stream flow. The model projects that if the climate of the past is cycled into the future, the mean flow could increase by 20%, and under the driest conditions, the mean could decrease by 50%. Under the latter's conditions, generation could plummet by 90% compared to the reference scenario (1981–2020). In this case, restricting ecological flow to 150 $m^3$/s would achieve a 207% improvement in hydropower generation. In addition, developing power plants 5 years ahead of their schedule would improve the normalized plant generation by 23%, while expanding reservoir volumes for planned power plants might yield no significant impact. It was also demonstrated that under arid conditions, stream flows could increase with increased radiative forcing.

To put the results in context, by 2020, 813 MW (64%) of the installed capacity on Uganda's national grid was located on the R. Nile and generated 3453 GWh (78%) [29]. With the planned commissioning of the Karuma hydropower plant in 2023, the production contribution of plants along the Nile will increase to 90%. This implies that the country's high reliance on the R. Nile for power production not only makes it highly susceptible to drought events but also questions the possibility of implementing the adaptive measures modeled in this study. Under the most severe conditions projected in this study, the current available grid capacity would reduce by 72% with a subsequent reduction of 83% in generation. It follows that the per capita consumption would drop from the current 110 kWh to 19 kWh. Restricting EFR to 150 $m^3$/s would only improve capacity by 10% and generation by 22% and this would result in a per capita consumption of 25 kWh, despite it being just a paltry 1% of the global average according to IEA [72]. It is envisaged that making alterations to EFR will be met with stern opposition from environmentalists and regulators and other geopolitical players (i.e., Egypt, Sudan, and Ethiopia) who have a large stake in the operations of the R. Nile. In essence, the most effective adaptive measure is also the least likely to be implemented. In the long term, effective resilience enhancement strategies would entail a combination of strategies. For example, hydropower plants could be developed with adjustable orifices for controlling EFRs and multiple turbines of varying capacities to be able to operate considerably at high efficiency with varying flows. In this case, low-capacity turbines could run at near-rated flow (and efficiency) during arid conditions.

Several generalizations were made in this study to aid computational expedience or limit the number of uncertainties introduced in the study. Future studies could explore different scenarios that take into consideration generation driven by electrical demand, hydropower production based on turbine efficiency curves, and the performance of the power system with the entire developed and planned generation mix. In addition, given the limitation of selected reanalysis datasets to model runoffs within watersheds with large reservoirs, future studies could explore downscaling and bias-correcting CMIP6 data for the Nile basin and utilizing data from GCMs that close the WBM. Other limitations of this study to be investigated in future works pertain to the inflexibility of the WEAP tool to model the natural flow of rivers emerging from lakes and the development of WEAP water balance models for L. Victoria and L. Kyoga, which do not rely on pseudo groundwater nodes to calibrate lake levels.

**Author Contributions:** Conceptualization, F.M; methodology, F.M.; software, F.M.; validation, F.M.; formal analysis, F.M.; investigation, F.M.; resources, F.M., and R.B.; data curation, F.M.; writing—original draft preparation, F.M.; writing—review and editing, F.M., T.B. and R.B.; visualization, F.M.; supervision, R.B., and T.B. All authors have read and agreed to the published version of the manuscript.

**Funding:** This research received no external funding.

**Institutional Review Board Statement:** Not applicable.

**Informed Consent Statement:** Not applicable.

**Data Availability Statement:** Not applicable.

**Acknowledgments:** Francis Mujjuni acknowledges the Ph.D. scholarship support from the Commonwealth Scholarship Commission in the UK.

**Conflicts of Interest:** The authors declare no conflict of interest.

## Abbreviations

| | |
|---|---|
| CLEWs | Climate Land Environment Water Systems |
| CMI | Climate Moisture Index |
| CMIP5 | Phase 5 Coupled Model Intercomparison Project |
| CMIP6 | Phase 6 Coupled Model Intercomparison Project |
| COD | Commercial Operation Date |
| DEM | Digital Elevation Model |
| ECMWF | European Centre for Medium-Range Weather Forecasts |
| E-flow | Environmental Flow |
| EFR | Environmental Flow Requirement |
| ERA | Electricity Regulatory Authority |
| ERA5 | Fifth Generation ECMWF Atmospheric Reanalysis of the Global Climate |
| FAO | Food and Agricultural Organisation |
| GCMs | General Circulation Models |
| HPP | Hydropower Plant |
| IEA | International Energy Agency |
| kWh | Kilowatt Hour |
| masl | Meters above Sea Level |
| MCM | millions of Cubic Meters |
| MW | Megawatt |
| nRMSE | Normalized Root Means Square Error |
| NSE | Nash–Sutcliffe Efficiency |
| PBIAS | Percentage Bias |
| $R^2$ | Coefficient of Determination |
| RCP | Representative Concentration Pathways |
| UBOS | Uganda Bureau of Statistics |
| USD | United States Dollars |
| WBM | Water Balance Model |
| WEAP | Water Evaluation and Planning |

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
