# Peer review of "Uganda’s Hydropower System Resilience to Extreme Climate Variability"

_climate, doi:10.3390/cli11090177_

Round 1

Reviewer 1 Report

The study is very interesting. My comments as follow: 

CLEWs, please at the abstract define the term to be clear to the reader

Introduction section. The authors maybe can refer to this study for drought (Spatiotemporal variability analysis of standardized precipitation indexed droughts using wavelet transform) and this study for hydropower(Prediction of hydropower generation via machine learning algorithms at three Gorges Dam, China)

"Figure 10: Climate Moisture Index for Various GCM-SSP Combinations", not clear 

"Figure 16: Comparison of R. Nile Observed Flow at L. Victoria Outflow for the period 2021-2022 with Selected Simula", more justification is needed. 

The authors should highlight the limitations of the current study in the Conclusion section.

Author Response

Comment 1: CLEWs, please at the abstract define the term to be clear to the reader

Response 1: Noted and effected.

Comment 2: In the Introduction section, the authors maybe can refer to this study for drought (https://doi.org/10.1016/j.jhydrol.2021.127299)

Response 2: We found the paper useful given that it presented alternative methods for characterising droughts. Under section 2.3.2 of our paper, this observation was referenced. 

Comment 3: "Figure 10: Climate Moisture Index for Various GCM-SSP Combinations", not clear.

Response 3: The Image has been rotated and it reads better. Alternatively, a separate image file is provided along with the manuscript.

Comment 4:  "Figure 16: Comparison of R. Nile Observed Flow at L. Victoria Outflow for the period 2021-2022 with Selected Simulation", more justification is needed.

Response 4: That particular section in general is asking the question of "how likely is it that the future climate would be similar to the projected extreme scenarios?". This is answered as "unlikely". So then we embark on testing the recorded 'future river flows' (Jan 2021- May 2022) with our projections within the same time span. Figure 16 is the graphical representation of the comparison. 

Comment 5:  The authors should highlight the limitations of the current study in the Conclusion section.

Response 5: This was effected.

Reviewer 2 Report

This manuscript quantifies the effects of extreme climate variability for various scenarios on hydropower production and proposes various compensation measures. The impact of climate change on hydropower production and the competing water use interests by other economic sectors is a current issue in many countries. 

The study entailed developing a water-balance model that accounts for the main hydrological features (lakes, and rivers), water demand sectors ecological flow requirements, and hydropower elements within the catchments of the large hydropower stations situated or planned along the R. Nile, with have a general lack of sufficient observational climate and hydrological data , therefore, the study leveraged data from General Circulation Models archived in the ERA5-Land and CMIP6 datasets.

The minor revision I request is a Recommendation, namely:

Figure 2 with the time series must be completed with another two graphs, namely the correlation of the average monthly flows between L. Victoria at Jinja Pier and the three sections with missing data (R. Victoria Nile at Mbulamuti, R. Kyoga Nile at Masindi Port, R. Kyoga Nile at Paraa). This must be done for the two WMO reference periods, namely 1961-1990,1991-2020, excluding unrealistic data. Obtaining a trend equation facilitates imputation of missing data. The equations for the three sections related to the two WMO reference periods will also be presented, a common aspect in hydrology. It would also be useful to perform the innovative trend analysis (ITA) method proposed by Åžen.

The effort put into this research is very high, especially since the authors faced a lack of data, which they solved by proposing some plausible scenarios. 

The article is worth publishing because it contains information useful to the readers of this journal.

I congratulate the authors for the clarity of the presentation of a complex analysis, thus facilitating the understanding of the article and I agree with the publication in this form.

Yours sincerely.

Author Response

Comment 1: Figure 2 with the time series must be completed with another two graphs, namely the correlation of the average monthly flows between L. Victoria at Jinja Pier and the three sections with missing data (R. Victoria Nile at Mbulamuti, R. Kyoga Nile at Masindi Port, R. Kyoga Nile at Paraa). This must be done for the two WMO reference periods, namely 1961-1990,1991-2020, excluding unrealistic data. Obtaining a trend equation facilitates imputation of missing data. The equations for the three sections related to the two WMO reference periods will also be presented, a common aspect in hydrology. It would also be useful to perform the innovative trend analysis (ITA) method proposed by Åžen.

Response 1: The graphs have been added to Figure 2 and regression equations were developed. However, we did not compute missing data given that in the comparison of modeled flows to observed flows, missing data points were not considered. In any case, we had sufficient data points at every gauging station that we did not need additional inferences to build a WEAP model.

Reviewer 3 Report

The manuscript entitled “Uganda’s Hydropower System Resilience to Extreme Climate Variability” is an original article presenting a particularly important study regarding the investigation of the study area’s hydropower system’s resilience under climate change scenarios and the effectiveness of adaptation measures for extreme aridity.

The manuscript provides valuable information for the study area, in terms of planning resilient infrastructure and establishing good practices in the frame of future water and hydropower management. It is well written, the methodological approach is very well documented and results are illustrated in general in a satisfactory way. Finally, the topic is within the scope of the journal Climate. However, I would like to suggest several amendments to the authors that could increase the quality of the manuscript and its suitability for publication:

- Figure 1: processes regarding water balance model should be presented in a more detailed way (and in accordance with parameters of paragraph 2.3.1 etc).

- Tables 3,12: check font size

- Lines 168-174, Figures 4-5: Regarding the simulation of the past period (1981-2020), information concerning the model input datasets, apart for runoff, i.e., for Precipitation, Wind, Relative Humidity etc. is based on reanalysis data. It is not clear if authors used for instance any Observations apart from the ERA5 etc datasets for any variable, and if so, did they evaluate the latter information? Finally, it is indeed a decisive issue that model calibration using observed levels/runoff timeseries (observations for the historic period) is based on input variables solely from reanalysis sources. Authors are advised to give more emphasis to this feature in their study, especially in recommendations for future work (l. 835-840), and also to mention any available previous study regarding statistical downscaling in the region; a fact that should be further discussed, after considering the spatial resolution of the datasets and the watersheds’ extend.

- Figures: modify left hanging and check figures captions’ alignment and line spacing

- Line 290 & paragraph 2.3.1 (WEAP tool): cite the software (Jack Sieber, Water Evaluation And Planning (WEAP) System, Software version: XXX, Stockholm Environment Institute, Somerville, MA, USA).

- Line 348: superscript for cbm is missing.

- Figure 9: This figure should be improved to increase readability.

Author Response

Comment 1:  Figure 1: processes regarding water balance model should be presented in a more detailed way (and in accordance with parameters of paragraph 2.3.1 etc).

Response 1: The water balance has 2 major elements; catchment characterization (both land classification and weather), and demand & supply dynamics. We believe that these two aspects are well represented in the inputs of the water balance model in Figure 1.

Comment 2: - Tables 3,12: check font size

Response 2: Font size for all tables adjusted to '10' in accordance to the journal template.

Comment 3: Lines 168-174, Figures 4-5: Regarding the simulation of the past period (1981-2020), information concerning the model input datasets, apart for runoff, i.e., for Precipitation, Wind, Relative Humidity etc. is based on reanalysis data. It is not clear if authors used for instance any Observations apart from the ERA5 etc datasets for any variable, and if so, did they evaluate the latter information? Finally, it is indeed a decisive issue that model calibration using observed levels/runoff timeseries (observations for the historic period) is based on input variables solely from reanalysis sources. Authors are advised to give more emphasis to this feature in their study, especially in recommendations for future work (l. 835-840), and also to mention any available previous study regarding statistical downscaling in the region; a fact that should be further discussed, after considering the spatial resolution of the datasets and the watersheds’ extend.

Response 3: Section 2.2 and specifically Tables 1-3 states the source of each data input. Section 2.2.3 clearly states that no observed weather data was used in the study. The reliance on reanalysis data was motivated by the lack of high spatial resolution data within the study area and the cumbersomeness in obtaining the meager existing observed data. As for your recommendation for exploring downscaling methods for future work, this had been done in Lines 558-559 and doubled down in Line 850. Regarding references for statistical downscaling, the authors could not find any within the study area. There is some work done by CORDEX (https://cordex.org/about/) but even so, this is on a continental (Africa) scale and predominantly for CMIP5. By the time we started our study, CORDEX had just introduced downscaled regional climate models for CMIP6 covering a few climate parameters.

Comment 4: Figures: modify left hanging and check figures captions’ alignment and line spacing

Response 4: Noted and effected.

Comment 5: Line 290 & paragraph 2.3.1 (WEAP tool): cite the software (Jack Sieber, Water Evaluation And Planning (WEAP) System, Software version: XXX, Stockholm Environment Institute, Somerville, MA, USA).

Response 5: See citation [46]

Comment 6: - Line 348: superscript for cbm is missing

Response 6: Fixed.

Comment 7: Figure 9: This figure should be improved to increase readability.

Response 7: Figure size increased.

Reviewer 4 Report

REVIEW OF UGANDA’S HYDROPOWER SYSTEM RESILIENCE TO EXTREME CLIMATE VARIABILITY

In my opinion the WEAP model as used by the authors in this study does not correctly represent the water balance of the Nile River watershed and consequently cannot be used to represent the changes in the hydrology of the watershed, and in the flows of the Nile River under possible future-climate scenarios.

The defect of the WEAP model as used in this paper is in the misrepresentation of the input and output terms which make up the water balance of Lake Victoria and its upland catchments which form the watershed under consideration.  

The authors cite the excellent two papers of Vanderkelen et al. (2018) on the modelling of Lake Victoria under current and future climate conditions.  In these two papers Vanderkelen et al. correctly state that the water balance of Lake Victoria is determined by two inputs (precipitation and inflow from upland areas) and two outputs (evaporation and outflow in the Nile River.

In the WEAP model the upland catchments (line 90) are treated as having one input (precipitation) and three outputs (evapotranspiration, groundwater recharge, and runoff).  For lakes (line 131) the two inputs are river inflows and precipitation and the three outputs are evaporation, river outflows, and losses to groundwater.

The major error present in the WEAP model as presented is that, for both upland catchments and lake catchments, recharge to groundwater is considered an output from the control volume. In fact, recharge to groundwater is an internal transfer within the control volume being analysed. Treating recharge to groundwater as an output is wrong - see the correct values presented by Vanderkelen et al.  For the lake catchments there is the additional error that recharge to groundwater through the lake bottom is either zero or upward into the lake in near-shore areas i.e., negative and not positive as stated in the paper.

The authors may have manipulated the data in some manner to end up with estimates of Nile River flows that correspond to measured values despite the misuse of recharge to groundwater as a model output.  Any such manipulation is specific to the existing pattern of climate variables, there can be no assurance that the hydrologic behaviour of the system will be correctly represented when changed climate variables are used with this physically incorrect model is used.

The correct way to proceed is to use hydrologic models that correctly represent the processes that determine streamflow. An example is the HydroGeoSphere model of coupled surface-subsurface hydrology as applied to the Laurentian Great Lakes (Cen et al.) 2020.   The physically correct modelling presented by Vanderkelen et al. (references 21 and 22).

I cannot recommend the present paper for publication because the WEAP model used to predict results from possible future scenarios is physically incorrect.

It might be possible to make the paper acceptable for publication if the authors acknowledged the defects in the WEAP mode, treated it as an entirely empirical procedure, and presented a table of comparison of annual Nile River flows as found in this study with annual Nile River flows as found by Vanderkelen et al. for similar scenarios (for example wet and dry scenarios).

As a note of explanation, I am concerned that every publication that deals with prediction of river flows gives proper attention to the interaction between surface and subsurface flows.   In the past some applications of the WEAP model have included findings that appear to treat surface flows as independent of groundwater. For example, a paper by Amisigo et al. that used WEAP concluded:

The water demands (municipal, hydropower, and agricultural) cannot be simultaneously met currently, or under any of the scenarios used, including the wet scenarios. This calls for an evaluation of groundwater as an additional source of water supply.

This conclusion suggests that groundwater is a separate flow system from surface water - a very dangerous error when making water-management decisions.

References

Amisigo B.A., McCluskey A. Swanson R. 2015  Modelling impact of climate change on water resources and agricultural demand in the Volta Basin and other basin systems in Ghana.  Sustainability 7(6) 6957-6975.

Chen J., Sudicky E.A.  et al.  2020  Towards a climate-driven simulation of coupled surface-subsurface hydrology at a continental scale: a Canadian Example  Canadian Water Resources Journal 45(1)  11-27.

Author Response

Comment 1: In my opinion the WEAP model as used by the authors in this study does not correctly represent the water balance of the Nile River watershed and consequently cannot be used to represent the changes in the hydrology of the watershed, and in the flows of the Nile River under possible future-climate scenarios.

Response 1: In this comment, no argument has been advanced and so there is nothing actionable but in response to comment 5 we offer a rebuttal to the reviewer's objections.

Comment 2: The defect of the WEAP model as used in this paper is in the misrepresentation of the input and output terms which make up the water balance of Lake Victoria and its upland catchments which form the watershed under consideration. 

Response 2: The authors employed a host of published literature and various formats of published WEAP Tutorials (text and videos) to undertake this work. The so-called ‘misrepresentation’ as alluded to by the reviewer is founded on their misinterpretation of our work as duly explained in response to comment 5.

Comment 3: The authors cite the excellent two papers of Vanderkelen et al. (2018) on the modelling of Lake Victoria under current and future climate conditions.  In these two papers Vanderkelen et al. correctly state that the water balance of Lake Victoria is determined by two inputs (precipitation and inflow from upland areas) and two outputs (evaporation and outflow in the Nile River.

Response 3: We direct the reviewer to tables 1-4 which state the data inputs into our study which is consistent with Vanderkelen et al [1], [2] In addition, we direct you to Vanderkelen et al [2] who cited a study that asserts the significance of subsurface flow. We also would like to inform the reviewer that they make false comparisons between our work and Vanderkelen’s. Vanderkelen was only interested in the water balance model for Lake Victoria, our work covers a basin 5 times bigger than that. In any case, Vanderkelen’s inflow model is essentially a precipitation-runoff regression model. In that respect, Vanderkelen only had precipitation as a single input but only categorized as lake precipitation and surface runoffs (inflow).

Comment 4: In the WEAP model the upland catchments (line 90) are treated as having one input (precipitation) and three outputs (evapotranspiration, groundwater recharge, and runoff).  For lakes (line 131) the two inputs are river inflows and precipitation and the three outputs are evaporation, river outflows, and losses to groundwater.

Response 4: In this description, you state rightly a typical WEAP model (see Yates et al., [3]). We make an exception for the two catchments with lakes by adding stream inflows as indicated in Table 4. Within WEAP upstream surface runoff, interflow, and baseflow are added onto the adjacent downstream catchment’s input. Again, this is consistent with Vanderkelen et al [1], [2] if we had only considered L. Victoria catchment as they did. Line 90 has been revised but the same argument had been stated in various places.

Comment 5: The major error present in the WEAP model as presented is that, for both upland catchments and lake catchments, recharge to groundwater is considered an output from the control volume. In fact, recharge to groundwater is an internal transfer within the control volume being analysed. Treating recharge to groundwater as an output is wrong - see the correct values presented by Vanderkelen et al.  For the lake catchments there is the additional error that recharge to groundwater through the lake bottom is either zero or upward into the lake in near-shore areas i.e., negative and not positive as stated in the paper.

Response 5: The characterization of our study as ‘wrong’ and Vanderkelen’s as ‘correct’ is simplistic. For instance, even though Vanderkelen’s work was groundbreaking and indeed has had so much utility, it has some questionable assumptions. Say, when they bias-corrected RCMs using reanalysis data (and yet referred to it observational data), or how wrong hydropower plant historical capacity was used, or the ‘sluicing’ of ‘excess’ flow from the lake and yet not considering it as part of river flow, or how lake levels were restricted (to a level of 10-13.5 m) even though we know that it can get lower and higher than this, etc. Again, we know that they did this not to obscure facts but rather to employ pragmatic assumptions to reproduce historical outflows using data that is highly biased. In fact, without bias correction (which is questionable in the way they did it), Vanderkelen failed to reproduce historically observed flows.

Groundwater nodes were added to every catchment and they all (except for the catchments with lakes) operated with an assumption that they recharged at 10% of surface runoffs. This is not a novel idea. Several researchers have employed similar methods such as [4], [5] but particularly a comprehensive hydrological study by the Ugandan government [6] proposes groundwater recharge within that range.

We did not consider groundwater within lakes as an output of the model but rather as a means for calibration. Once calibration was done for the historical levels, then the derived stream flows for the ‘future’ or for different scenarios, entirely depended on the input model data. That is, the outflow only changed to reflect the GCM data input. Vanderkelen et al., [1], [2] went behind this problem by; (1) first, finding GCM that best approximated the lake levels, (ii) restricting water levels within a narrow range and (iii) bias-correcting the ‘other’ GCMs to predict the future. On our part, we used CMIP6 data (rather than CMIP5 used by Vanderkelen). CMIP6 had been billed to be more accurate than CMIP5, so our study was in part to test this hypothesis. We found out that such claims were overly exaggerated. So faced with massive data uncertainties within CMIP6, lack of observational data to bias-correct GCMs, and the lack of flexibility of WEAP to model the natural flow of the river which has a lake as its source, we employed pseudo groundwater nodes (within catchments with lakes) to calibrate lake volumes. We reckon that with the use of downscaled and bias-corrected data, this groundwater node might not be needed, a hypothesis we seek to explore in the next stage of our work.

The method we implemented in this study to calibrate the WEAP Model is not entirely novel. Hughes et al’s commentary on the Pitman model indicates that the water balance model can be calibrated “with either interflow or groundwater outflow as the dominant process” [7]. In addition, Dehghanipour et al., [8] calibrated their WEAP Model by directly controlling (fixing) groundwater recharge through an estimation of pumping requirements. The novelty in our work is that we place these groundwater nodes with the catchments with the highest uncertainties and propose piecewise linear regression models for effecting the calibration process.

Comment 6: The authors may have manipulated the data in some manner to end up with estimates of Nile River flows that correspond to measured values despite the misuse of recharge to groundwater as a model output.  Any such manipulation is specific to the existing pattern of climate variables, there can be no assurance that the hydrologic behavior of the system will be correctly represented when changed climate variables are used with this physically incorrect model is used.

Response 6: The accusations of manipulating the data are quite unfortunate, seeing that the reviewer completely misunderstood our use of groundwater nodes. Every model in the literature on Lake Victoria seeks to reproduce historical levels by directly controlling the water levels. In fact, you can argue that every water balance model for L. Victoria is circular in nature. Our contribution to this realization is the use of pseudo groundwater nodes during the calibration process. Once the calibration is effected, change in the lake levels can only be from applied climate data. This is attested by our study of how cycling historical climate data, in time, the model reproduces past river outflow trends. We have added a bit more context to our methods. See lines 511-521.

Comment 7: The correct way to proceed is to use hydrologic models that correctly represent the processes that determine streamflow. An example is the HydroGeoSphere model of coupled surface-subsurface hydrology as applied to the Laurentian Great Lakes (Cen et al.) 2020.   The physically correct modeling presented by Vanderkelen et al.

Response 7: We spent nearly 20 pages of this study relaying data and tools that would equip any researcher to reproduce our work. We know not one study that has labored to publish both its calibration and validation methods as we have done. We have disclosed all assumptions we employed and the limitations within them. We also compared our study against others including Vanderkelen et al.

Comment 8: It might be possible to make the paper acceptable for publication if the authors acknowledged the defects in the WEAP mode, treated it as an entirely empirical procedure and presented a table of comparison of annual Nile River flows as found in this study with annual Nile River flows as found by Vanderkelen et al. for similar scenarios (for example wet and dry scenarios).

Response 8: We have validated our results against 4 gauging stations unlike in any study we know. Secondly, we discussed our results and how they compare with other studies. In any case, what the reviewer asks, we have done, in part, in the results section, and above all our WEAP model managed to reproduce historical observations for 40 years and demonstrated that if historical climate data is cycled, it is capable of reproducing the same trend for the future. We also demonstrated in our work that we were well acquainted with  Vanderkelen’s work and indeed cited them a number of times both in the methods and results section. See the context added to our calibration method in Lines 511-521.

Comment 9: The water demands (municipal, hydropower, and agricultural) cannot be simultaneously met currently, or under any of the scenarios used, including the wet scenarios. This calls for an evaluation of groundwater as an additional source of water supply.

Response 9: We have no idea where we made any such inference or this is a conclusion from prior studies.

Comment 10: This conclusion suggests that groundwater is a separate flow system from surface water - a very dangerous error when making water-management decisions.

Response 10: Again, this is a wrong inference but enough has already been explained on this matter.

References

[1]         I. Vanderkelen, N. P. M. Van Lipzig, and W. Thiery, “Modelling the water balance of Lake Victoria (East Africa)-Part 2: Future projections,” Hydrol Earth Syst Sci, vol. 22, no. 10, pp. 5527–5549, 2018, doi: 10.5194/hess-22-5527-2018.

[2]         I. Vanderkelen, N. P. M. Van Lipzig, and W. Thiery, “Modelling the water balance of Lake Victoria (East Africa)-Part 1: Observational analysis,” Hydrol Earth Syst Sci, vol. 22, no. 10, pp. 5509–5525, 2018, doi: 10.5194/hess-22-5509-2018.

[3]         D. Yates, J. Sieber, D. Purkey, and A. Huber-Lee, “WEAP21—A Demand-, Priority-, and Preference-Driven Water Planning Model,” https://doi.org/10.1080/02508060508691893, vol. 30, no. 4, pp. 487–500, Dec. 2009, doi: 10.1080/02508060508691893.

[4]         V. Sridharan, E. P. Ramos, C. Taliotis, M. Howells, P. Basudde, and I. V Kinhonhi, “Vulnerability of Uganda’s Electricity Sector to Climate Change: An Integrated Systems Analysis,” in Leal Filho W. (eds) Handbook of Climate Change Resilience, Springer, Charm, 2019, pp. 1–30. doi: https://doi.org/10.1007/978-3-319-71025-9_45-2.

[5]         M. Khaki and J. Awange, “The 2019–2020 Rise in Lake Victoria Monitored from Space: Exploiting the State-of-the-Art GRACE-FO and the Newly Released ERA-5 Reanalysis Products,” Sensors, vol. 21, no. 4304, 2021, doi: https://doi.org/10.3390/s21134304.

[6]         Directorate of Water Resource Management, “Consolidated Hydrological Year Book for Uganda 1978-2014,” Kampala-Uganda, 2014.

[7]         D. A. Hughes, S. Mantel, and F. Farinosi, “Assessing development and climate variability impacts on water resources in the Zambezi River basin: Initial model calibration, uncertainty issues and performance,” J Hydrol Reg Stud, vol. 32, p. 100765, 2020, doi: 10.1016/j.ejrh.2020.100765.

[8]         A. H. Dehghanipour, B. Zahabiyoun, G. Schoups, and H. Babazadeh, “A WEAP-MODFLOW surface water-groundwater model for the irrigated Miyandoab plain, Urmia lake basin, Iran: Multi-objective calibration and quantification of historical drought impacts,” Agric Water Manag, vol. 223, no. February, p. 105704, 2019, doi: 10.1016/j.agwat.2019.105704.

Round 2

Reviewer 4 Report

I do nThe control volume has an upperot find the paper suitable for publication because the authors do not appear to be aware of the correct procedure for defining inputs and outputs for water balance calculations for watersheds.

The correct definition for the control volume of a watershed is as follows:

(1) The control volume has as its  upper surface the contact surface between the atmosphere and the earth's surface (including any surfaces such as buildings and vegetation attached to the earth's surface) . The boundary of the upper surface is the topographic watershed boundary that separates surface flow paths leading to outflow channels for surface flows (if there are one or more) from flow paths leading to flow channels that are outside the watershed .

(2) The sides of the control volume are vertical planes extending from the location of the watershed boundary on the earth's surface downward to a layer of rock or soil that restricts vertical water flow to insignificant rates.

(3) The bottom surface of the control volume is the aformentioned low permeability layer.

For a corrected defined watershed control volume the Inputs are (i) precipitation, (ii) lateral groundwater flow into the control volume through the sides of the control volume (iii) any water brought into the watershed by constructed conveyance (pumped water in pipes or channels.

The Outputs of the watershed control volume are (i) evapotranpiration, (ii) streamflow exiting the watershed as channel flow at the outlet  locations on the watershed boundary (iii) lateral groundwater flow leaving the watershed volume through the sides of the control volume (iv) any water exported from the watershed by pumping or other conveyance.

For large watersheds such as the Nile Watershed the only input of significant magnitude is precipitation and the only outputs of significant size are evapotranspiration and streamflow.

As I stated in my original review the transfer of water from the surface of the watershed to groundwater is an internal transfer within the control volume and it is a major error to describe this transfer as a watershed output.   A related error is to call the transfer of water from the surface to groundwater as a loss.  This water is not lost to the flow system but stays within the watershed flow system and contributes to either streamflow or evapotranspiration

A second separate error is the authors'  assertion that the direction of groundwater movement in the bottom of large lakes is from the lake to groundwater. This is incorrect. For large lakes there is a small region of lake bottom near shore with groundwater flow from the groundwater system to the lake. For most of the lake bottom there is no groundwater flow in either direction.

Until the authors modify the paper to remove the error in identifying groundwater recharge as a loss and adjust watershed outputs to not include groundwater recharge as a watershed output there is no point in my conducting further  review of  any further submission of a revised paper.